# Real-time signal processing via chemical reactions for a microfluidic molecular communication system

Vivien Walter [1,4], Dadi Bi [1,4], Ali Salehi-Reyhani [2,3] & Yansha Deng [1] ✉

Signal processing over the molecular domain is critical for analysing, modifying, and synthesising chemical signals in molecular communication systems. However, the lack of chemical signal processing blocks and the wide use of electronic devices to process electrical signals in existing molecular communication platforms can hardly meet the biocompatible, non-invasive, and size-miniaturised requirements of applications in various fields, e.g., medicine, biology, and environment sciences. To tackle this, here we design and construct a liquid-based microfluidic molecular communication platform for performing chemical concentration signal processing and digital signal transmission over distances. By specifically designing chemical reactions and microfluidic geometry, the transmitter of our platform is capable of shaping the emitted signals, and the receiver is able to threshold, amplify, and detect the chemical signals after propagation. By encoding bit information into the concentration of sodium hydroxide, we demonstrate that our platform can achieve molecular signal modulation and demodulation functionalities, and reliably transmit text messages over long distances. This platform is further optimised to maximise data rate while minimising communication error. The presented methodology for real-time chemical signal processing can enable the implementation of signal processing units in biological settings and then unleash its potential for interdisciplinary applications.

Communication technologies for transmitting information over distances have been revolutionising the way humans, sensors, and robots communicate. Since the invention of the telephone by Bell in 1876, significant progress has been made over a century in both wired and wireless communications, with major innovations including transmitting optical signals over optical fibre in a confined cable and electromagnetic wave signals over the air in an unbounded environment. Different from wired and wireless communications that are designed for typical human-to-human, human-to-machine, or machine-to-machine macroscale applications (e.g., mobile communications), molecular communication (MC) was proposed for the first time to explore the exchange of information using chemical molecules, mimicking the communication processes found in nature, such as the transmission of pheromones in the air among individuals and the hormone signalling in fluid media inside the human body[1,2]. MC research holds immense potential in emerging applications where traditional wired or wireless communication would be unsafe or infeasible, such as intrabody biosensing[3], smart drug delivery[4,5], and explosive gas monitoring[6].

Over the past decade, the main focus of MC research was to theoretically characterise and analyse an MC system using tools and mechanisms from communication engineering, and advancement

[1]Department of Engineering, King's College London, London WC2R 2LS, UK. [2]Department of Surgery and Cancer, Imperial College London, London W12 0HS, UK. [3]Institute for Molecular Science and Engineering, Imperial College London, London SW7 2AZ, UK. [4]These authors contributed equally: Vivien Walter and Dadi Bi. ✉e-mail: yansha.deng@kcl.ac.uk

includes modelling the biophysics of transceiver and propagation[7-9], quantifying communication capacity[10,11], evaluating and optimising communication performance[12-14]. Aligning with the ongoing theoretical research efforts, researchers also realised the importance of developing experimental MC platforms in validating the proposed theoretical models and facilitating practical MC-based applications[15,16]. However, the development of an experimental MC platform is complex as it requires multidisciplinary expertise (e.g., communication engineering, chemistry, biology, and physics)[2], and to date only a few gas and liquid-based MC platforms have been successfully prototyped[17-36].

Like any communication system, an MC platform can be generally divided into three fundamental parts: a transmitter, a propagation channel, and a receiver. So far, the existing experimental MC platforms focused on the propagation through bounded and unbounded spaces, and their main objective was to demonstrate the feasibility of exchanging information via various chemicals, which includes not only the molecules found in nature, e.g., volatile organic compounds[17-21], acids[22,23], protons[24], glucose[25], DNA molecules[26], and sodium chloride[27], but also the carriers designed in the lab, e.g., droplets and bubbles[28], artificial magnetic nanoparticles[29-31], fluorophores[32,33], colour pigments[34], and carbon quantum dots[35,36]. In existing MC platforms, their MC transceivers are interfaced with macroscale instruments (spray[17,18,33], pump[20,23,25,27-29,31,34-36], pH sensor[22-24], light detector[32,35,36]) for the conversion between electrical signals and chemical signals, i.e., release and observe information molecules. It is important to note that the electrical devices here are integrated with these macroscale instruments to perform signal processing functions (e.g., encoding-decoding[22] and detection algorithms[18,22,23,25,27]) based solely on electrical bit signals, with the aim of ensuring successful information transmission after propagation. Although it is acceptable to use electronic devices for signal processing when focusing on the information exchange based on chemical molecules, the utilisation of electronic devices for signal processing functions can hardly meet the biocompatible, non-invasive, and size-miniaturised requirements of most microscale/nanoscale biochemical applications, e.g., tissue engineering, targeted drug delivery, and immune system enhancement[37]. Thus, it is important to shift from electrical signal processing to chemical signal processing and design a family of pure chemical concentration signal processing (e.g., pulse shaping and thresholding) blocks that operate in real-time for signal generation, detection, amplification, etc. Nevertheless, the experimental implementation of these basic signal processing functions over chemical signals, especially at the microscale/nanoscale, has been ignored so far.

Inspired by nature where cells rely on a complex molecular network to process biochemical signals via reacting molecules[38,39], chemical reactions emerge as an efficient way to construct a wide range of biochemical function blocks for sensing, computation, actuation, etc[40-44]. This motivates us to utilise chemical reactions to address the challenge of performing signal processing functions over the molecular domain. Considering that chemical reactions are more likely to occur in the liquid phase[45,46], here we develop a liquid-based microfluidic molecular communication (MIMIC) platform. Specifically, through a joint design of a selection of chemical reactions and of a finely tuned microfluidic geometry for the transceiver, our MC transmitter is capable of shaping the transmitted signals, and the MC receiver is capable of detecting an incoming signal exceeding a pre-defined threshold and amplifying it to a user-defined level. Compared to the previously reported platforms, the signal processing functions in our MIMIC platform are realised over the molecular domain and are free of electronic devices, only using macroscale external devices to interface with our transmitter and receiver for chemical injection and quantitative measurement and to provide complete control for the performance evaluation.

To ensure that the mechanism of our platform can be easily understood by different disciplines, we choose a selection of universally known pH-based reactions, based on sodium hydroxide (NaOH) and hydrochloric acid (HCl), to demonstrate and quantify the feasibility of our general chemical reaction design for electronic-free chemical signal processing functions. While these reactions might not lead to a specific application in medicine or industry, their universality can perfectly illustrate that the design principles behind our platform can be applied to any kind of reaction that would be relevant to a specific application. Using these reactions, we experimentally demonstrate that our designed MIMIC platform can reliably transmit bit signals with a low bit error rate (BER), even at the highest transmission speeds achievable by our hardware and over long distances, testing tubing up to 25 m long. To facilitate accurate and reproducible measurement of our platform's performance, we have also developed a Python-based software with a graphical user interface (GUI) that entirely automates signal generation through multi-pump synchronisation and control, real-time chemical signal recording, and data visualisation. The interplay between flow chemistry and microfluidics established in this work can lead to other designs of pure chemical concentration signal processing blocks for arithmetic operation, logic computing, signal transformation, etc. Benefiting from the modularity of our platform, our MC transmitter and receiver are expected to operate individually for microfluidic-based applications by replacing the acid and base with application-specific chemicals and designing the corresponding chemical-reaction-based signal processing functions. This includes but is not limited to medicine and biology applications, where the transmitter could efficiently communicate with cells and their membranes (e.g., drug delivery, DNA sensing), and the receiver could decode and read chemical signals emitted by an organism (e.g., cell culture monitoring, detection of illness-specific biomarkers).

## Results
### MIMIC platform design
In this study, two critical challenges need to be addressed for the development of a liquid-based MC platform that relies on chemical reactions for signal processing: (i) a flow chemistry challenge, jointly designing the microfluidic geometry and chemical reactions to regulate the occurrence of reactions, which enables the process of time-varying concentration signals in the molecular domain and timely communication, and (ii) a communication engineering challenge, prototyping microfluidic MC transmitter and receiver with molecular signal modulation and demodulation functionalities via the above flow chemistry designs, where the communication efficiency can be controlled and optimised.

To overcome these challenges, we designed and developed the MIMIC platform to explore signal processing capabilities via chemical reactions to manipulate concentration signals and facilitate reliable communication for MC. Our MIMIC platform consists of a transmitter that generates and emits signal Y with the information encoded to its concentration, a microfluidic propagation channel, and a receiver that decodes and detects incoming signal Y (Fig. 1a). In our platform, we selected an ultraviolet-visible (UV-Vis) spectrometer as the detector at the receiver due to its capacity to detect and identify a large selection of molecules.

To address the first challenge, we elaborately designed a series of chemical reactions inside the transceiver to achieve three main signal processing functions: 1) signal shaping, 2) signal thresholding, and 3) signal amplification and detection. To segregate these reactions at different locations of the transceiver, we built the whole system using configurable microfluidic tubing.

With the injection of signal Y at the transmitter inlet during a given time interval, we can adjust the duration of the signal Y emitted at the transmitter outlet by injecting a signal suppressor P within the same time interval. The signal suppressor P completely consumes Y and the duration of emitted signal Y is controlled by the timing of the

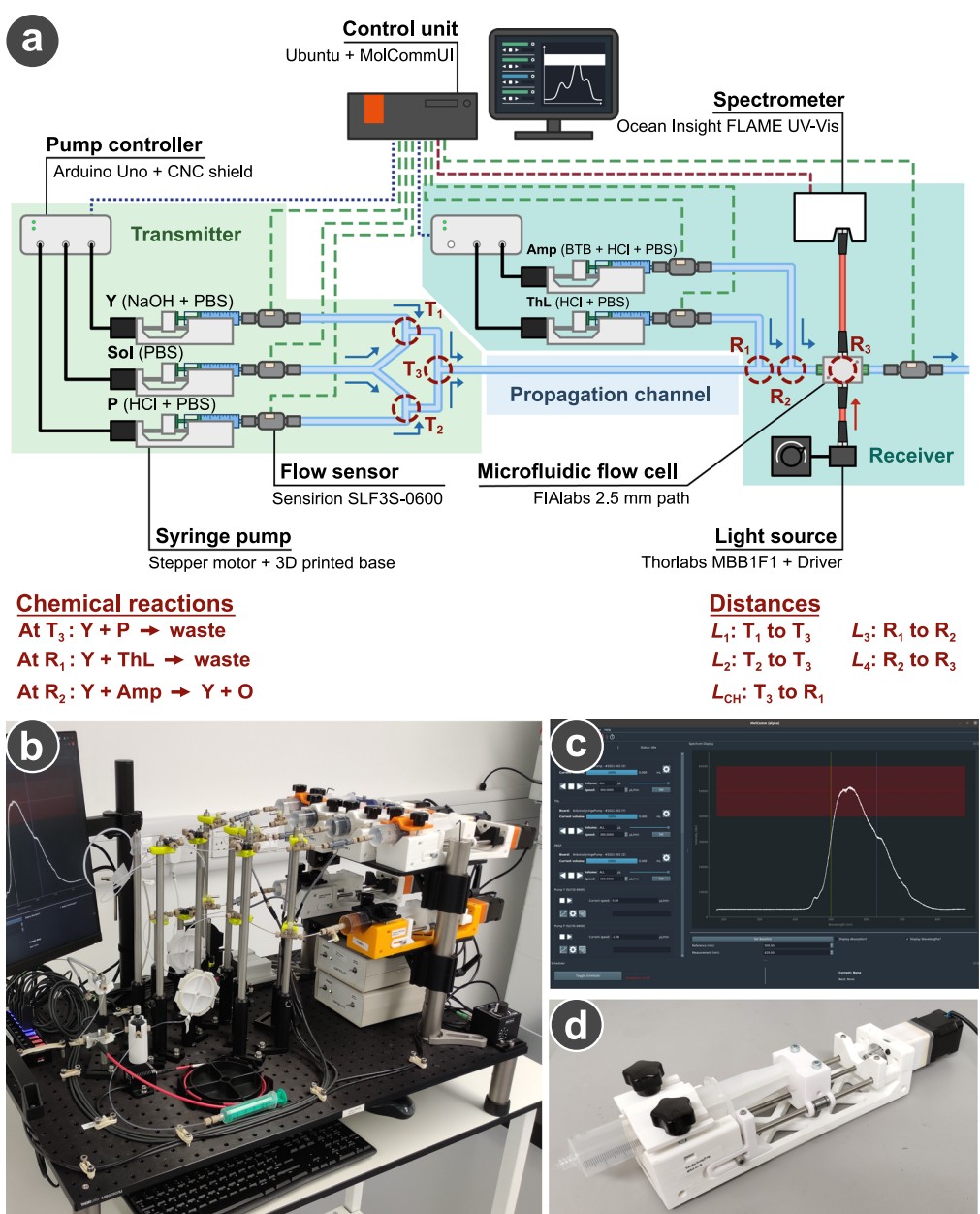

**Fig. 1 | MIMIC platform design. a** Architecture of our MIMIC platform, presenting all the elements used to build it. The symbols Y, P, Amp, ThL, and Sol represent the different chemical solutions injected in the platform. $T_1$ to $T_3$ correspond to the locations in the transmitter where different injected chemical solutions mix together. $R_1$ and $R_2$ are the locations in the receiver where different injected chemical solutions mix together. $R_3$ is the location at which the absorbance of a solution is measured. The variables $L_1$ to $L_4$ and $L_{CH}$ are the different tubing lengths used in this work. The three communication parts making the platform have been highlighted in different coloured zones. The blue, green, and red non-continuous lines represent the cable connections to exchange commands, flow rate data, and absorbance data, respectively. The make and models are provided for each piece of equipment. **b** Picture of the platform built in this paper, along with **(c)** a screenshot of the Python control software (full screenshot available in Supplementary Fig. 13) and **(d)** a picture of one of our 3D-printed syringe pumps.

interaction between Y and P, either by delaying the injection of P or its propagation. After propagation and upon reception of the signal Y at the receiver, we introduced thresholding and amplification reactions for signal detection. We first designed a thresholding operation to mitigate the noise and intersymbol interference (ISI) introduced by the propagation channel through the reaction between Y and a signal thresholder ThL. This reaction effectively consumed all the signal Y below a user-defined concentration level of ThL. Since only the concentration above the threshold will contribute to the detected signal, this allows for a precise selection of the minimum concentration of Y required for the following signal detection. To amplify the remaining signal Y to a detectable level, we then devised the amplification

reaction between Y and a signal amplifier Amp to generate a receiver output O visible for the spectrometer. Through this chemical interaction at the receiver, the visualisation of the reception of signal Y was allowed, and the concentration of the output O could be amplified to a desired constant as long as the concentration of Y exceeds the predefined concentration of ThL.

Using chemical reactions in microfluidic setups to process the signal imposes a critical implementation requirement: each chemical reaction should be completed before the solution reaches the next important location. In a microfluidic setup, it is possible to control the time it takes for a solution to go from one point to another by adjusting the tubing length separating these two points. Given a tubing of length

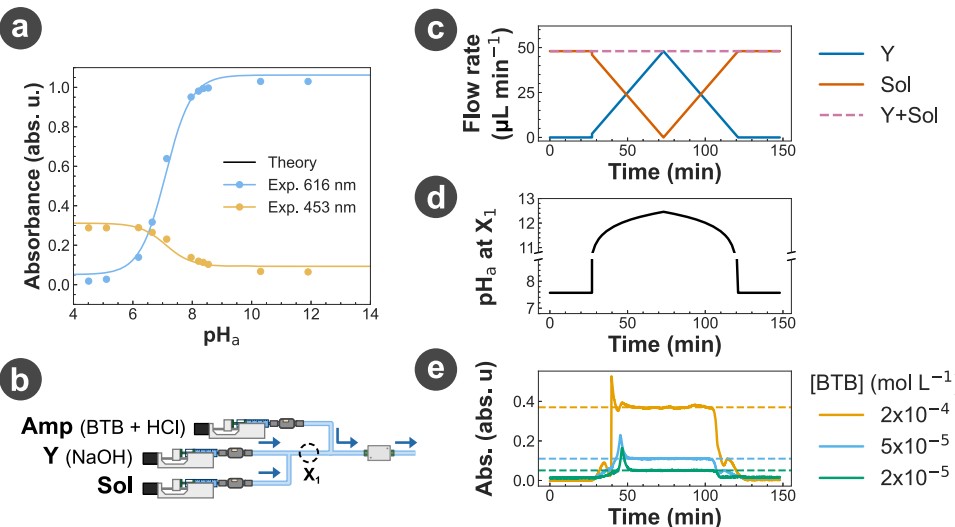

**Fig. 2 | Signal amplification and detection. a** Evolution of the absorbance of a BTB solution at 453 and 616 nm with the actual pH (pH$_a$) of the solution. Experimental data are compared with the theoretical predictions obtained using Supplementary equation (7). The concentration of BTB [BTB] was 0.2 mmol L$^{-1}$. **c–e** Response of the amplification reaction to a non-square incoming signal. **c** Flow rate profiles of the pumps injecting Y and Sol. **d** Analytical prediction of the pH$_a$ at location X$_1$ shown in

(**b**) based on the calibration in standard glassware (Supplementary note 4). The symbols Y, Amp, and Sol represent the different chemical solutions injected in the platform. **e** Measurement of the absorbance at 616 nm of the solution using the spectrometer for different concentrations of BTB [BTB] in Amp. The profiles in (**c**) and (**d**) have been shifted by the observation time for ease of comparison with (**e**). pH$_Y$ = 10.8 and pH$_{Amp}$ = 5.3.

$L$ and inner cross-section area $S$, the time $T$ it takes for the solution to propagate is calculated as $T = LS/Q$, where $Q$ is the flow rate of the solution. Therefore, by changing the length $L$ of the tubing we can directly set at which time $T$ the solution will reach the end of the tubing. To ensure that a chemical reaction is complete before the solution reaches the end of the tubing length, the corresponding minimum tubing length $L_{min}$ that will allow for the completion of the reaction should be calculated and used as a condition on the tubing length in the design of our MIMIC platform. In particular, $L_{min}$ can be calculated for any chemical reaction via $L_{min} = QT_R/S$, where $T_R$ is the reaction time of that reaction. Note that the value of the reaction time $T_R$ can be directly measured in the microfluidic setup by measuring a quantity related to the reaction yield (Supplementary Note 3).

To realise and experimentally validate these three signal processing functions, we chose NaOH as Y, HCl as P and ThL, and bromothymol blue (BTB) as Amp. By doing so, the transmitted information was encoded in the concentration of H$^+$ or OH$^-$ ions, i.e., the pH of a solution. With NaOH as Y and BTB as the main component of Amp, the interaction of Y with Amp will result in a change in solution colour, which can be easily quantified by the spectrometer. To provide a baseline and to perform dilutions, a solvent Sol, made of phosphate-buffered saline (PBS) solution, is injected into the platform. The Sol pump can also be used to keep the total output flow rate of the platform constant.

## Signal amplification and detection

We first verify the behaviour of the amplification reaction between Y and Amp at the receiver, which is designed to amplify the incoming signal Y and produce an output signal O visible to the UV-Vis spectrometer via

$$Y + Amp \leftrightarrow Y + O. \tag{1}$$

In order to achieve signal amplification at the receiver of our platform just before the signal is converted by the UV-Vis spectrometer into an electrical signal, our amplification reaction at the receiver should follow two design principles: 1) in the presence of Y, the concentration value of O should be a constant value determined by the user and not

influenced by the concentration of input Y, while 2) the produced output signal O should be visible and quantifiable to the spectrometer. It should also follow the implementation requirement that the concentration of O should only be determined by this reaction rather than influenced by the hydrodynamics of the platform. Following these two design principles and fulfilling this requirement will make the chemical Amp the signal amplifier in our design. Moreover, since in our MIMIC platform the signal Y, with the NaOH solution as the main component, is transparent and cannot be detected at the receiver using UV-Vis spectroscopy, the chemical Amp acts as both the signal amplifier and signal detector at the same time. Despite having both properties, we will only be referring in text to Amp as the signal amplifier.

Knowing that the information is encoded in the pH of solution Y, we selected a pH-sensitive dye for Amp to address the first design principle. This is because pH-sensitive dyes can effectively convert a change in the pH of the solution into a change in the colour spectrum of a solution that can be detected by the spectrometer. However, the colour of a pH-sensitive dye is determined not only by the pH of the solution but also by the local dye concentration which can fluctuate in a microfluidic setup. These fluctuations, caused by cases such as when the injection pumps are turned on or off, violate the implementation requirement. We resolved this issue by selecting a specific pH-sensitive dye for Amp, i.e., BTB, because BTB can be characterised by two forms: an acid form, i.e., HBTB, with a maximum absorbance at 453 nm and a base form, i.e., BTB$^-$, with a maximum absorbance at 616 nm[47,48]. The existence of these two forms with each featured by a maximum absorbance wavelength enabled us to develop a mathematical framework to convert the colour spectrum measurement of the spectrometer into an indirect measurement of the pH of the solution that is independent of the dye concentration, thus satisfying that implementation requirement (see Fig. 2a, Methods, and Supplementary note 2.3).

The second design principle is also fulfilled by using BTB as the main component of the Amp solution, i.e., generating the output O with a constant concentration in the presence of signal Y. Through the catalytic reaction of HBTB with NaOH, the evolution of the concentration of output BTB$^-$ with the input concentration of NaOH is

non-linear and can be mathematically described by

$$[BTB^-] = [BTB]_{Tot} \frac{10^{14} K_C [NaOH]}{1 + 10^{14} K_C [NaOH]}, \qquad (2)$$

with $[BTB]_{Tot}$ as the total concentration of BTB, in both acid and base forms, and $K_C$ as the equilibrium constant of reaction (1) (Supplementary Note 2.1). This sigmoid evolution implies that outside a narrow transition range, a large variation in [NaOH] will only result in a small variation of $[BTB^-]$, resulting in the concentration evolution satisfying the second design principle of the amplification reaction. The conversion of the concentration of $BTB^-$ into the absorbance of the BTB solution is then described by the Beer-Lambert law (Supplementary equation (1)). By combining the non-linear conversion of the concentration of NaOH into the concentration of $BTB^-$ with the linear conversion of the concentration of $BTB^-$ into the absorbance of the BTB solution, we obtain a non-linear conversion of NaOH into the BTB absorbance. Through our proposed mathematical transformation of the BTB absorbance into a revised pH value ($pH_r$), the sigmoid evolution of $[BTB^-]$ results in a $pH_r$ evolution that spans between a low and a high pH limits. In this way, the actual pH values ($pH_a$), proportional to [NaOH], measured outside these limits are converted to either the low or the high revised pH limits (Fig. 2a). Using this absorbance-to-pH conversion based on the acid and base forms in BTB, non-square signals (e.g., triangular or Gaussian signals) can be clipped into square-like signals.

As our platform can only allow us to directly control the time-varying input flow rate but not the chemical concentration, we generated non-square concentration signals through an adjustment of flow rates using the setup in Fig. 2b. Signal Y was injected following a triangular profile in flow rate while the output flow rate was kept constant (Fig. 2c) which would lead to a Gaussian profile in concentration and $pH_a$ (Fig. 2d) before the reaction with Amp at location $X_1$ (Fig. 2b). As expected, the output $pH_r$ measured using the spectrometer after the reaction with Amp would be a square signal (Fig. 2e) instead of the triangular or Gaussian signal, verifying that using BTB for the Amp solution satisfied the second design principle. We also demonstrated in Fig. 2e that using BTB for the Amp solution gave us the ability to control and amplify the receiver output of our platform to a desired level by adjusting the concentration of BTB [BTB]. However, it is noted that using BTB as the Amp can only and strictly be valid if the relationship between the absorbance measured and the BTB concentration in the Amp solution is linear, but does not hold for concentrations of BTB outside this linear range (Supplementary note 2.2). Experimental results in Fig. 2 demonstrated that the pH-sensitive dye nature and the unique feature of BTB addressed both the design principles as well as the implementation requirement, making BTB a perfect candidate for our Amp solution.

## Signal thresholding

To reduce the noise and ISI introduced by the propagation channel, it is essential to define a minimum concentration of Y, called the activation concentration, at which the amplification reaction (1) can be activated, thus generating the chemical output O that can be measured at the spectrometer. In order to remove the signal Y that is below a specific threshold before signal amplification, a thresholding reaction is also integrated into the receiver of our MIMIC platform before the amplification reaction can occur, where the concentration of received signal Y is lowered by the solution ThL via the designed reaction

$$Y + ThL \rightarrow \phi, \qquad (3)$$

where $\phi$ is a type of molecule irrelevant for signal transmission. Through this reaction, the activation concentration of the amplification reaction can be controlled by the selection of the concentration of ThL.

With the NaOH solution selected as the transmitted signal Y, we chose an HCl solution as the signal thresholder ThL and verified the effect of ThL on a solution of Y in traditional glassware. In Fig. 3a, we measured the minimum concentration of Y required to trigger the generation of O ($BTB^-$ form, blue) from Amp (HBTB form, yellow), before and after the addition of ThL. We observed that the $pH_a$ at which the BTB activation is triggered was shifted by the addition of ThL, meaning that a higher concentration of Y is required to activate BTB. It is noted that the shift of the BTB activation also depends on the initial $pH_a$ of the BTB solution.

Following the reception of signal Y, its reaction with ThL would result in a decrease in signal Y's amplitude. If the concentration of ThL is too high compared to the concentration of Y, the signal Y will be completely suppressed. We demonstrated how the selection of the ThL concentration (Fig. 3e) results in a selection of the minimum Y concentration (Fig. 3d) that can be detected at the spectrometer using the experimental setup in Fig. 3b. In this experiment, the flow rate of solution Y was increased while the output flow rate was kept constant (Fig. 3c), thus resulting in a gradual increase of $pH_a$ (Fig. 3d). Then by injecting ThL solutions with different concentrations of HCl, we were able to select the minimum $pH_a$ of solution Y that toggles the generation of output O. The flow rate of Y was then gradually decreased to demonstrate that the detected signal would disappear at the same $pH_a$ as the one it appeared at (Fig. 3e). The experimental data in Fig. 3a-e demonstrate that, given a chemical signal Y such as NaOH, it is possible to design a thresholding reaction that can select the minimum concentration required to trigger the signal detection, thus allowing the user to mitigate noise and ISI without the requirement of external electronic devices.

## Signal suppression

The transmitter of our MIMIC platform is able to suppress the input signal Y over time before emission at the output of the transmitter through the reaction with the chemical P, which is

$$Y + P \rightarrow \phi. \qquad (4)$$

For a continuous injection of Y and P, a delayed arrival of P after Y at the junction, where both chemicals are expected to mix and react, would shape Y into a square-like pulse by the reaction, thus generating a pulse-like signal at the output of the transmitter.

To effectively suppress the signal Y made of a NaOH solution, we selected an HCl solution for the signal suppressor P. Indeed, if the HCl solution is concentrated enough, P will consume all the NaOH from the Y solution and no signal will be emitted by the transmitter. We verified our chemical design by observing that a continuous emission of a signal Y at the transmitter output would be completely suppressed following the injection of P without a change in the injection of Y (Fig. 3f, g). If the injection of P is made after a $T_d$ delay in time, the output signal displays a square-like shape as predicted, with a width $T_w$ approximately equal to the delay $T_d$ (Fig. 3f, g).

## Message transmission

With individual chemical-reaction-based microfluidic circuits designed and experimentally verified for signal shaping, thresholding, amplification, and detection to address the first challenge, we then addressed the second challenge by integrating these specific chemical-reaction-based microfluidic circuits into a MIMIC prototype to realize the time-varying molecular signal modulation and demodulation for text message transmission over a distance on the order of metres.

The communication was divided into intervals with a duration $T_b$ during which the transmitter was scheduled to convey one-bit information per interval. For binary bit-1, the syringe pumps were turned on to inject chemical signal Y and solution P from the beginning of the bit interval until time $\alpha T_b$, with $\alpha$ as the duty cycle ($\alpha \leq 1$), and were turned

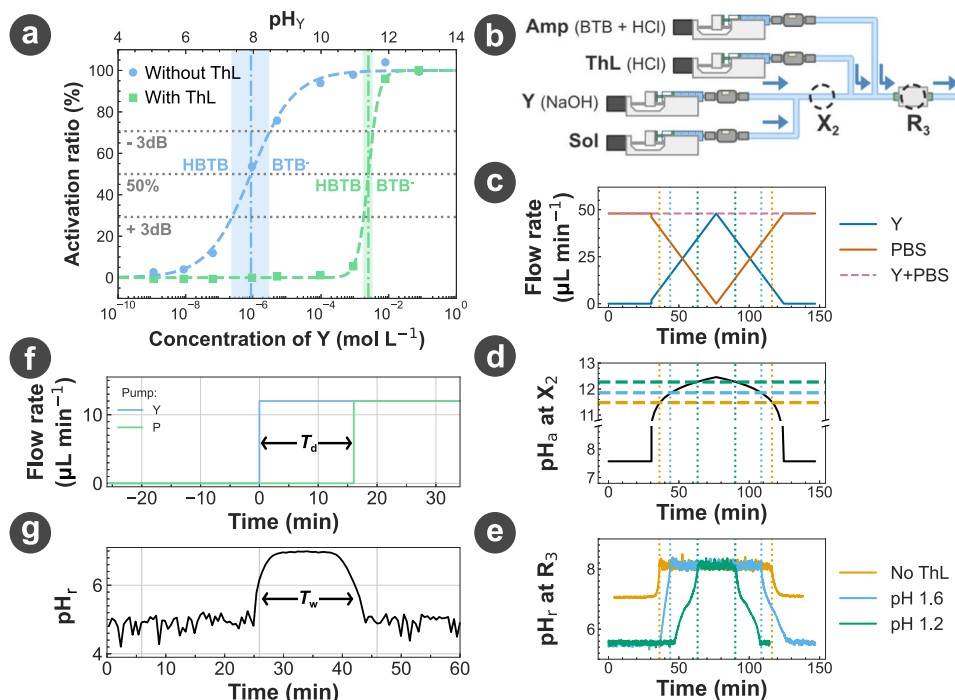

**Fig. 3 | Signal thresholding and suppression. a** Activation profile of the BTB with varying concentrations of Y (i.e., pH$_Y$), measured before and after a 1:4 dilution of Y +Amp in a ThL solution. The activation ratio is calculated as the relative level of absorbance between the absorbance of a pure HBTB solution (activation ratio 0%) and the absorbance of a pure BTB$^-$ solution (activation ratio 100%). The switch values between the acid and base forms and the transition range are shown on the graph, where the transition range is characterised using a 3 dB threshold ($\approx 71\%$) from both the low and high level absorbance. The dashed lines are sigmoid fit of the experimental data used as visual guidelines. **b–e** Response of the thresholding reaction to a non-square incoming signal measured using the experimental setup shown in (**b**). **c** Flow rate profiles of the pumps injecting Y and Sol. **d** Analytical prediction of the pH at location X$_2$ (Supplementary note 4). **e** Measurement of the pH of the solution at location R$_3$ using the spectrometer for different pH values of the solution ThL. For the case of No ThL, ThL was replaced by Sol. **f, g** Effect of the injection of solution P after a time $T_d$ following the injection of solution Y on the transmitter output signal Y measured using the spectrometer. $T_w$ is the measured width of the resulting pulse-like signal. All pH and concentrations for Y, P, ThL, Amp, and Sol are provided in the Methods.

off for the remaining time of the bit interval acting as a guard interval. For binary bit-0, the syringe pumps containing Y and P were turned off during the whole bit interval (Fig. 4a, b).

With the objective of transmitting the message "Hi", our MIMIC prototype generated the corresponding binary sequence "1001000 1101001" (ASCII). Two extra bits "11" were also concatenated at the beginning of the information bit sequence to indicate the start of a message and the duration of a single bit $T_b$ for the receiver. Figure 4a demonstrated that the transmitted message was successfully demodulated at the receiver with error-free transmission, due to the long bit interval and small flow rate ensuring completeness of all reactions in tubing and a stable flow rate output from syringe pumps. This not only validates the design of our MIMIC platform for delivering information via chemical molecules but also enables us to further investigate the communication efficiency of the system.

### High speed and long distance communication

To explore the limits of the communication performance of our MIMIC platform, we investigated the communication efficiency for transmissions at the highest speeds achievable by our system and over a long distance. In our platform, the flow speed is mainly limited by the minimum and maximum flow rates of syringe pumps, decided by our selected stepper motors. This flow rate range can be further reduced by the increase of back pressure in the tubing at high speeds and the effect of diffusion and advection.

To examine the impact of the increasing back pressure and signal dispersion with the increase of the length of the propagation channel, we explored the limits of our MIMIC platform by first replacing the propagation channel with the longest channel available in our lab, i.e., $L_{CH} = 25$ m. This tubing length conveniently

corresponds to more than 10 times the average longest straight distance in the human body (head to toe). We further assessed the communication performance by gradually increasing all the injection flow rates by a speed factor of $N$ to their limit governed by the pump hardware, up to 200 µL min$^{-1}$ for a single pump. Through transmitting multiple bit-1 signals (Fig. 4c), our results show that our MIMIC platform can modulate and demodulate an increased number of bits over the same time duration using narrower pulses, thus allowing for $N$ times increase in the data rate. Interestingly, we did not observe any significant speed-induced distortion of the signal shape experimentally, even at the highest speed achievable by our hardware (see Fig. 4d for the pulse width $T_w$ and Supplementary Note 6.5 for other pulse shape descriptions). These experiments demonstrate the capacity of our MIMIC platform to achieve high data rates under the constraints set by its design, although the highest possible data rates of our system are found lower than others previously reported in the literature[31,32]. However, to achieve a specific high data rate, a careful selection and setting of the transmitted pulse width $T_e$ is needed to ensure that the received output pulse width $T_w$ remains wide enough to be detected.

### Waveform design

Although increasing the communication speed by $N$ times leads to a higher data rate, the shortening of the width of a transmitted signal $T_e$ makes the signal more susceptible to noise and interference during propagation, especially over a 25-metre long communication distance. It is therefore essential to control $T_e$ (Fig. 4b).

When generating a bit-1 in our MIMIC platform design, the width of the pulse emitted by the transmitter $T_e$ is initially controlled by the injection duration $\alpha T_b$ of Y such as $T_e = \alpha T_b$, and will directly impact the

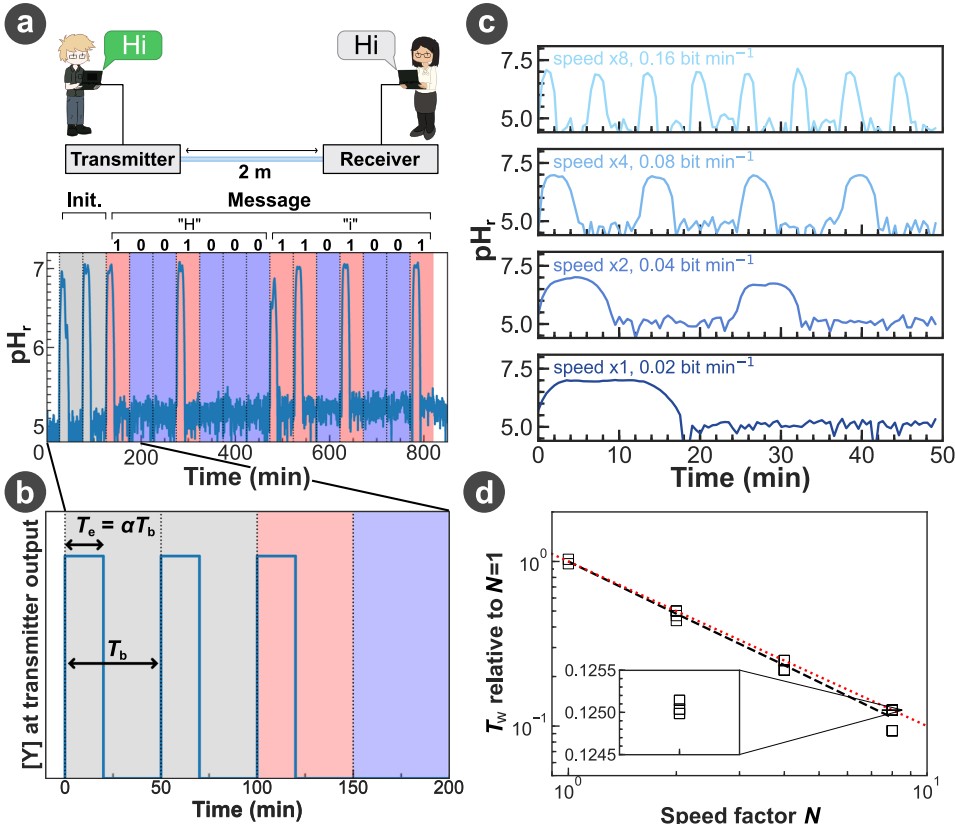

**Fig. 4 | Communication performance. a** Transmission of the message "Hi" (ASCII: "1001000 1101001") using the bit encoding illustrated in **b**. Bits "11", shown in grey, are concatenated at the beginning to signal the start of a message and define the duration of the bit interval $T_b$ and the duty cycle $\alpha$. **c** Comparison of demodulated signals transmitted using the platform at different bit rates, achieved by applying different transmission speed factors $N$ ranging from 1 to 8. **d** Evolution of the demodulated signal width $T_w$ with the speed factor $N$. The black dashed line is the average evolution of the data presented, and the red dotted line is the proportional evolution with $1/N$. Solution composition and flow rates are given in the Methods and Supplementary Note 6.4.

width of the pulse at the output of the receiver $T_w$. It is important to note that in the design used in this work, we were only able to measure experimentally $T_w$ but not $T_e$. Upon use of the signal suppressor P to shape the signal as illustrated through reaction (4) in Fig. 3f, g, $T_e$ is no longer defined by $\alpha T_b$ but instead by $T_d$, i.e., the delay between the injection times of Y and P (Fig. 5a, c). However, setting $T_e$ and thus $T_w$ via the difference in injection time means that $T_e$ and $T_w$ are both defined through the control software of the platform, an external electronic device. In order to fully address the second development challenge of our MIMIC platform, the width of the emitted pulse $T_e$ should only be defined by the flow chemistry setup.

To remove the reliance on the electronic device to determine the emitted pulse width $T_e$, the injection time of Y and P were first synchronised ($T_d = 0$) by the software, then the propagation channel of P was lengthened by $L_d$ to delay the arrival of P at the mixing junction with Y (Fig. 5b). Our experiments demonstrated that the exact same signal shaping obtained by the controlled injection delay of P at the electronic device can be achieved by designing the length difference $L_d$ of microfluidic geometry, showing an equivalent effect of $T_d$ and $L_d$ (Fig. 5c, d).

To achieve the same output signal width $T_w$, Fig. 5e presents the direct mapping between the required channel length difference $L_d$ and the injection time difference $T_d$. This mapping can also be extended to compute an estimation of the final output signal width $T_w$, as a function of both $L_d$ and $T_d$ (Fig. 5f), achieving full control of the signal shape at the output of the transmitter using both the control software and the platform geometry.

## Communication performance optimisation

Among the three reactions occurring in our MIMIC platform, we investigated how the pulse generation performed by the reaction between Y and P at the transmitter following the geometry design in Fig. 5b can impact the communication reliability of the platform. We specifically examined the impact of the duty cycle $\alpha$ used in the encoding of the bit-1 on BER. As $\alpha$ directly controls the width of the transmitted signal $T_e$ and thus the width of the output signal $T_w$, two issues can arise depending on whether the value of $\alpha$ is too high or too low. First, if $\alpha$ is too low, $T_w$ can be so narrow that it could result in a loss of the signal amplitude and lead to the wrong detection of bit-1 signals as bit-0. Second, if $\alpha$ is too high, the long duration of a bit-1 signal can make it disperse to adjacent bits and likely cause the wrong detection of bit-0 signals as bit-1, especially when the transmitted bit sequence includes a pattern of "101". Fig. 6a showcases the second issue for various $\alpha$ values above 0.6, thus highlighting that high $\alpha$ values may result in an increase in communication errors. The complete decoded bit sequences are shown in Supplementary Fig. 11, and the typical bit errors observed during the measurement are presented in Supplementary Fig. 12.

In the case of high $\alpha$ values, the use of P in our design as a signal suppressor can cause signal distortion. Because of the late arrival of P as well the long pulse duration of P, the signal Y for consecutive bits can be consumed twice by P over two-bit durations thus resulting in a smaller width $T_{e,2}$ in the second bit as in Fig. 6b. The narrower pulse width $T_{w,2}$ for the second bit (Fig. 6b) will be hard to decode as bit-1 due to its low amplitude.

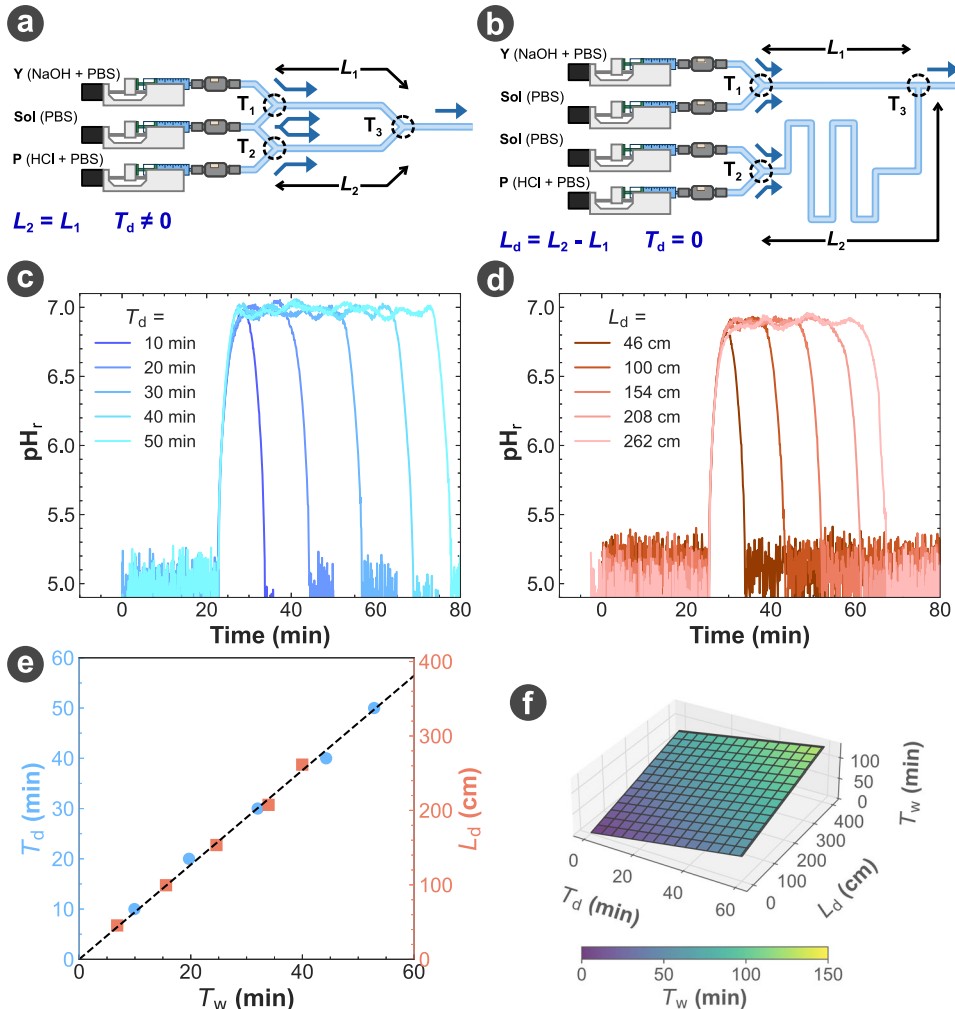

**Fig. 5 | Waveform control. a, b** Geometries used for the transmitter setup to control the output signal width $T_w$ by **(c)** changing the injection delay $T_d$, or by **(d)** modifying the length difference $L_d$, respectively. **e** Comparison of the measurements in **(c)** and **(d)** on the width of the measured signal $T_w$, shown as blue and orange points, respectively. Both datasets were fitted using a linear function and the resulting fits were aligned (dashed line). **f** 3D projection of the models found in **(e)**. For **(d)**, $L_1 = 5$ cm. The pH and concentrations for Y, P, ThL, Amp, and Sol are provided in the Methods.

To avoid this signal distortion caused by the signal Y reacting two times with the signal suppressor P over consecutive bit-1 signals, the bit interval $T_b$ should be larger or equal to the sum of the width of the transmitted pulse $T_e$ (equivalent to $T_w$ in our model) and the emission time $\alpha T_b$, hence

$$T_b \geq T_e + \alpha T_b. \tag{5}$$

Otherwise, the next signal Y will be emitted while P is still being injected. Equation (5) can be rewritten as a condition on $T_b$, such as

$$T_b \geq \frac{T_e}{1 - \alpha}, \tag{6}$$

where $T_e$ can be set either directly by the injection time difference $T_d$ in the control software or induced by the platform geometry and the selection of a path difference $L_d$ between the inlets of Y and P greater than 0. Figure 6c experimentally demonstrates the existence of the $T_{w,2}$ distortion regime (D) and non-distortion regime (ND) below and above the optimal bit interval $T_b^\star$, which aligns with the minimum $T_b^\star = T_e/(1 - \alpha)$ to be set to avoid distortion, as theoretically presented in equation (6).

With the optimisation goal of maximising the data rate while minimising the communication error, the bit interval $T_b$ has to be designed to be small enough to allow more bits transmitted, and large enough to prevent signal distortion. This leads to the optimal value of $T_b^\star = T_e/(1 - \alpha)$, which we verified experimentally (Fig. 6d).

## Discussion

The biological nature of chemical signalling makes MC a promising methodology for interdisciplinary applications ranging from personal healthcare to environmental science. Although a few gas/liquid-based MC platforms have been developed and demonstrated the feasibility of transmitting information via various types of molecules[17,22,24–29,32,34,35], the chemical dimension of these platforms is strictly restricted to the use of molecules as a signal carrier, and all the necessary signal processing functions of these platforms are entirely performed using external electronic devices. This lack of chemical concentration signal processing would hinder the adoption of these platforms to practical applications. Motivated by the previous theoretical MC works that revealed the chemical signal processing capabilities of chemical reactions[40–44], we report in this paper the experimental implementation of a MIMIC platform that achieves real-time signal shaping, thresholding, amplification, and detection functions for chemical concentration signals in the molecular domain.

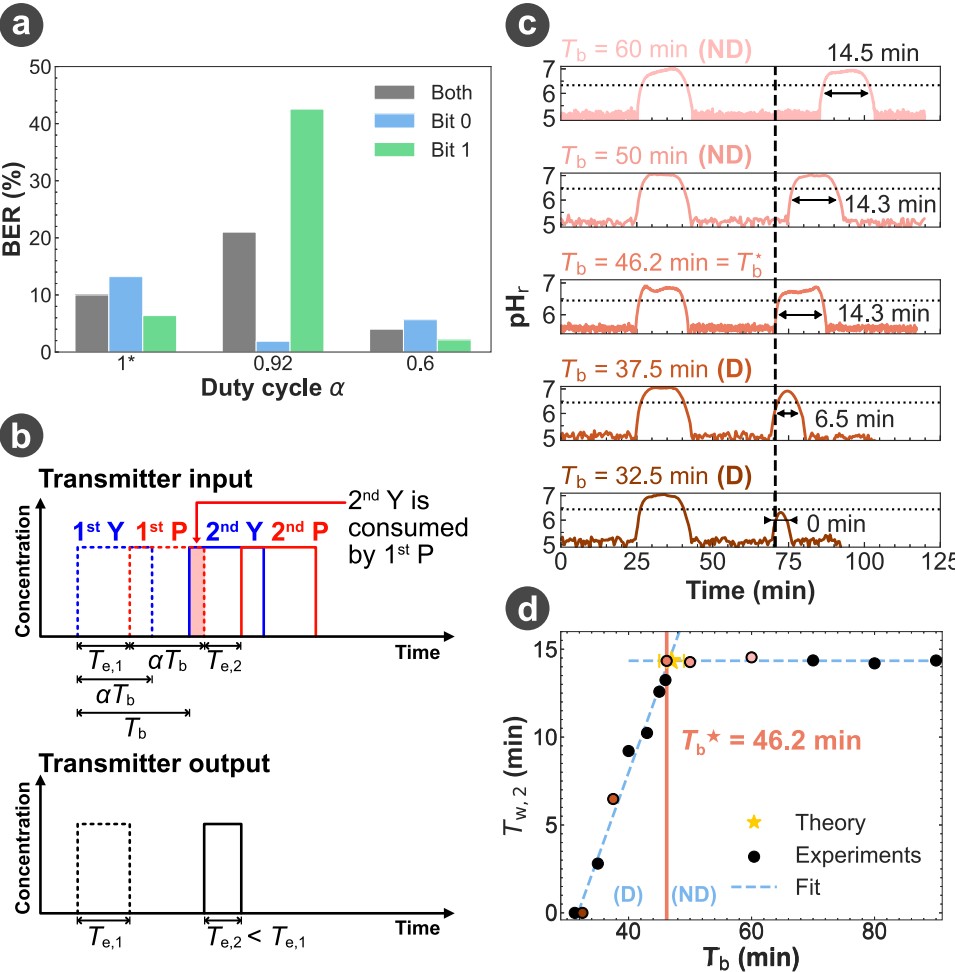

**Fig. 6 | Bit interval optimisation. a** BER measured over the same random 100-bit sequence (53 bit-0 and 47 bit-1) for 3 different duty cycle $\alpha$. For $\alpha = 1$, P was not injected into the platform. The speed factor was set to $N = 8$ and $L_{CH} = 25$ m. **b** Distortion regime (D) caused by the injection of P during a bit emission on the shape of the following bit if the bit interval is too short ($T_b < T_e + \alpha T_b$). **c** Reception of two bit-1 emitted with different $T_b$ and $\alpha$, while $\alpha T_b$ was fixed to 30 min. The measured widths $T_{w,2}$ of the output signals for the 2nd bit are shown on the graph. The horizontal dotted line represents the detection threshold used for pulse identification, and the vertical dashed line represents the observed time below which the second Y is consumed by the first P. **d** Evolution of $T_{w,2}$ with $T_b$, measured in the same conditions as for (**c**). The solid orange line highlights the minimum $T_b^{\star}$ observed experimentally and separates $T_b$ in the ND and D regimes described in (**b**). The colored data points correspond to the measurement extracted from the curves shown in (**c**). For all experiments, the setup and parameters used were the ones of Fig. 5b.

A key aspect of performing signal processing in the molecular domain is the versatility of these functions, which theoretically can be applied to any molecular signal carrier and to any form of encoding electrical bit signals to chemical signals. There are two methods to encode information into chemical signals: 1) intrinsic encoding, where the information is stored in changes of the physico-chemical properties of the individual molecules (e.g., DNA structure[26], stereo-isomerisation of fluorophores[49,50]), and 2) extrinsic encoding, where the interpretation of the information is based on the collective properties of the chemical solution (e.g., concentration of the solution[22,23], magnetic susceptibility[29]). Correspondingly, processing chemical concentration signals will therefore either modify the physico-chemical properties of a molecule or modify the composition of a solution. To present the proof-of-concept of our design and illustrate clearly the versatility of our design, the experiments presented in this work for our MIMIC platform rely on an extrinsic encoding of the information into the concentration of a base solution, NaOH. Thus, all chemical reactions presented in this work process NaOH concentration signals based either on modifying the composition of a NaOH solution (i.e., acid-base reactions for signal shaping and thresholding) or on the change of the physico-chemical properties of a pH indicator by the presence or absence of NaOH (i.e., signal amplification and detection). Although other MC studies already investigated encoding signals in acid and base solutions[22,23], our MIMIC platform uses reactions to process signals in real-time, whereas other platforms left all the processing tasks to computers and micro-controllers.

The advancement and versatility of our platform are also reflected in the selection of the hardware. The previously published studies on acid and base encoding in MC all used a pH-meter to extract information from concentration variations, while our MIMIC platform used a UV-Vis spectrometer. This choice was made to better illustrate the versatility of our design: while a pH-meter is a convenient solution to quantify the properties of acid-base solutions, it cannot be easily applied to other chemical reactions due to its specificity. By contrast, while a UV-Vis spectrometer is not capable of directly detecting changes in acidity, we have successfully demonstrated how chemical reactions can convert an invisible chemical signal into a visible chemical signal. The capability of our chemical-reaction-based signal processing functions to change the nature of the signal carrier brings a versatility that allows us to adapt the signal to the detector type at the receiver, and combined with the natural versatility of the UV-Vis spectrometer proves the versatility of our MIMIC platform.

We evaluated the communication performance, and we explored the limits of our platform in communication, specifically to transmit and recover a text message over a long distance and at the highest speeds achievable by our hardware, which imposes higher noise and interference levels. In such conditions, our MIMIC platform was able to maintain a satisfying reliability, even with a distance as long as 25 m. We have also demonstrated that the use of our proposed chemical-reaction-based signal processing blocks allowed us to control the communication efficiency and the BER. However, while our platform successfully transmitted information at the highest transmission speed achievable by our platform with chemical signal processing capability, this speed is lower than the MC platform without chemical signal processing capability[31,32]. This is because the highest transmission speed of our platform is limited by the reaction times of all the chemical signal processing functions, as well as the maximum injection speed of our 3D-printed pumps. It is important to note that higher data rates can be achieved by adapting our MIMIC platform with more powerful pumps and other kinds of faster chemical reactions.

It is important to highlight the fact that all the experiments and equipment presented in this work are used as a strict proof-of-concept platform to validate the feasibility of chemical signal processing using the universally known acid and base-based chemical reactions as an example. However, our MIMIC platform is not limited to these reactions and hardware, and practical applications of our MIMIC platform and its processing functions, e.g., in biology, medicine, or environment sciences, can be achieved by adapting the chemical reactions and the platform accordingly. The versatility of our design makes it easy to replace the chemicals and the reactions used in this work with corresponding molecules specifically dedicated to the sought application. For instance, with a fluorescent dye as the signal carrier, signal suppression (i.e., used for signal shaping and thresholding) can be achieved with a black hole quencher, while signal amplification and detection can be achieved using another fluorescent probe by Förster resonance energy transfer (FRET)[51,52]. Moreover, in order to validate our design and test its performance, several electronic devices are still needed to inject the solutions (i.e., syringe pumps) and to convert the chemical signal back into a measurable and quantifiable electrical signal (i.e., the UV-Vis spectrometer). Of course, an actual application of the design of our platform would require the setup to be adapted at either the transmitter and/or receiver specific to that application. For example, to read the chemical signals strictly encoded in the acidity of a water solution extracted from the environment, the UV-Vis spectrometer could be replaced with a pH-meter. Meanwhile, to communicate with a receptor cell, one can directly remove any form of electronic sensor.

For all potential applications, our low-cost plug-and-play platform allows for rapid testing of iterative design, optimisation, and validation of application-related chemical reactions and microfluidic geometry. After validation, compact, light, and portable lab-on-a-chip devices can be produced in one go by adapting the syringe pumps to micropumps and replacing the tubing and junctions with microfluidic chips. In this perspective, by proposing a versatile, low-cost, and adaptable design capable of replacing all electronic devices with biocompatible and non-invasive signal processing functions in the molecular domain, our MIMIC platform is a major step towards the multidisciplinary applications promised by MC, e.g., in medicine, biology, chemistry, and environment sciences.

## Methods
### MIMIC platform fabrication
The three parts composing the MIMIC platform, i.e., the transmitter, propagation channel, and receiver, were all assembled using standard microfluidic tubing (either PEEK or PTFE, OD 1/16 inch, ID 0.75 mm) and PEEK Tee junctions (OD 1/16 inch, ID 0.050 inch, IDEX Material Processing Technologies, USA) to connect the inlets with the outlets of

each part in the geometries presented in Figs. 1a, 2b, 3b, 5a, and 5b. PEEK and PTFE were both chosen for their compatibility with a wide range of chemicals, as well as the flexibility and space requirement between two junctions in the platform. PTFE was due to its high flexibility in tight spaces or coiled to achieve extra-long lengths (e.g., 25 m), while PEEK was used due to its resistance under mechanical stress essential in certain locations of the platform.

Chemical solutions were injected at the inlets of the transmitter and receiver using up to six syringe pumps that were developed and 3D-printed. The design of the pump was inspired by one of the Poseidon Pumps[53–55] with significant modifications: specific clamps and vices were adapted to fit the syringe model used in this work (3-part 50 mL syringe, Terumo, Japan), and additional parts were made to constrain and stabilise the rotation of the main axle (Fig. 1d). Each pump was also fitted with a 1:51 gearbox (Stepperonline Planetary gearbox, purchased on Amazon UK) to adapt the flow rate range to 1–200 μL min⁻¹, and regulated by a flow meter (SLF3S-0600F, Sensirion, Switzerland) to stabilise the flow rate via a feedback loop control. The outlet of the transmitter was directly connected to the inlet of the receiver via the propagation channel.

The outlet of the receiver was connected to a SLF3S-0600F flow meter to monitor the stability of the flow rate in the platform, and to a flow cell (Z-type 2.5 mm pathlength, stainless steel, FIALabs instruments, USA) to measure the absorbance of the solution over time. This measurement was achieved by connecting a light source (470–850 nm fibre-coupled LED, Thorlabs, USA) and a UV-Vis spectrometer (FLAME-T-UV-Vis spectrometer, 3648 px, Ocean Insight, USA) to the flow cell.

### Software
The MIMIC platform is controlled by our graphic user interface (Fig. 1c), implemented in Python 3 (the Python Software Foundation, US). Briefly, the software translates the input signals (e.g., constant, pulses, triangular waves) into a series of commands to control the flow rate of the pumps of the platform over time. The software then sends the commands to the Arduino controlling the pumps via serial communication, while collecting data from the flow meters and the spectrometer at the same time. To ensure the stability of the flow rate of the pumps during the injection, the data extracted from the flow meters are used by our software to apply a proportional-integral-derivative (PID) controller to the pumps.

The Arduino controls the pumps by implementing the commands sent by the software via a custom script written using the AccelStepper Arduino library (https://www.airspayce.com/mikem/arduino/AccelStepper/index.html). Data are extracted from the flow meters via serial communication, and from the spectrometer using the python-seabreeze library (https://github.com/ap--/python-seabreeze).

### Chemical solution preparation
Bromothymol blue (BTB, dye content 95%, ACS reagent grade), Hydrochloric acid (HCL, 37%, ACS reagent grade), Sodium Hydroxide (NaOH, ≥98%, pellets, ACS reagent grade), Phosphate-Buffered Saline (PBS, tablets), and Ethanol (≥ 99.8%, GC grade) were all purchased from Merck (ex-Sigma Aldrich, Germany). Ultra-pure water was obtained from an Elga Purelab DV35. The BTB stock solution was prepared by dissolving 20 mg of the commercial BTB powder in a solution of 16 mL NaOH at 0.02 mol L⁻¹, 8 mL Ethanol, and 30.4 mL ultra-pure water.

The solutions Y, P, ThL, and Sol were all prepared from fresh PBS solution, and their pH values were adjusted by adding a small volume of HCl or NaOH (< 1 % of the total volume of PBS) until the desired pH values were reached. The pH of the solution was measured and adjusted using a phenomenal PH 1100 L pH-meter (VWR, USA). The solution Amp at pH 5.3 was prepared by mixing 1 volume of the BTB stock solution with 3 volumes of PBS solution. Unless otherwise stated, the concentration of BTB used was 0.2 mmol L⁻¹, and the compositions

**Table 1 | List of the different solutions injected into the MIMIC platform, with their respective labels, pH values, compositions, and the corresponding dominant species found in solutions**

| Label | pH | Chemicals | Dominant species |
|-------|------|-------------|---------------------------|
| Y | 12.5 | NaOH, PBS | $OH^-$, $PO_4^{3-}$ |
| P | 1.5 | HCl, PBS | $H^+$, $H_3PO_4$ |
| ThL | 1.6 | HCl, PBS | $H^+$, $H_3PO_4$ |
| Amp | 5.3 | BTB, HCl, PBS | HBTB, $H^+$, $H_2PO_4^-$ |
| Sol | 7.58 | PBS | $H_2PO_4^-$ |

and pH values of the solutions injected into the MIMIC platform are summarised in Table 1. All experiments were performed in traditional glassware instead of a microfluidic setup and used a PBS solution at pH 7.58 as solvent.

### Flow rate signal generation

The injection of the signal Y by a syringe pump was programmed to follow either a triangular or a square profile in flow rate. For the experiments in Figs. 2c and 3c, the flow rate of chemical signal Y followed a triangular distribution. Instead of directly using a triangular profile, it was approximated by a staircase signal that allows the PID controller to have enough time to reach desired flow rates, and the staircase signal is in the form of

$$Q_Y(t) = \sum_{i=1}^{24} 2i\{u[t - 2(i-1)] - u[t - 2i]\}$$
$$+ \sum_{i=24}^{47} (96 - 2i)\{u[t - 2(i-1)] - u[t - 2i]\}, \qquad (7)$$

where $u[t]$ is the Heaviside step function with respect to time $t$.

For the experiments in Figs. 4, 5, and 6, the flow rate of input signal Y was defined by a square signal, which can be expressed as

$$Q_Y(t) = Q_{Y_0} \sum_{i=0}^{I} s_i\{u[t - iT_b] - u[t - iT_b - \alpha T_b]\}, \qquad (8)$$

where $Q_{Y_0} = 12\ \mu L\ min^{-1}$, $s_i$ represents the transmitted bit information (i.e., $s_i = 0$ or 1), $I$ is the number of transmitted bits, $T_b$ is the bit interval, and $\alpha$ is the fraction of the bit interval where the pump is turned on to represent bit-1.

At the transmitter side, to control the duration of the transmitted signal, solution P can be injected at a time $T_d$ later than Y by software programming, thus the flow rate of input solution P can be expressed as $Q_P(t) = Q_Y(t - T_d)$. The flow rate of the pump that injects solution Sol is determined by whether the transmitter setup is a 3-pump (Fig. 5a) or a 4-pump configuration (Fig. 5b). For the 3-pump configuration, the flow rate of solution Sol can be described as $Q_{Sol}(t) = Q_{TX} - [Q_Y(t) + Q_P(t)]$; while for the 4-pump configuration that contains two Sol pumps, the flow rates are $Q_{SolY}(t) = Q_{TX}/2 - Q_Y(t)$ and $Q_{SolP}(t) = Q_{TX}/2 - Q_P(t)$, respectively. Here, $Q_{TX}$ is the total output flow rate of the transmitter. At the receiver side, the flow rates of ThL and Amp pumps were kept constant all the time. Unless otherwise stated, $Q_{TX}$ was set to be 48 $\mu L\ min^{-1}$, and the flow rate of each pump at the receiver was set to be 16 $\mu L\ min^{-1}$. Thus, the total flow rate of the output of the whole platform was 80 $\mu L\ min^{-1}$.

### Microfluidic experiment protocol

The experiments of generating and transmitting chemical signals to different tubing setups followed the same generic protocol. The MIMIC platform was first implemented to the architectures presented in Figs. 2b, 3b, or Fig. 1a with the transmitter set to the geometry

presented in either Fig. 5a or b. Before connecting pumps to the platform, the setup was primed by manually injecting all chemical solutions at their corresponding inlets. Then, the pumps were connected to the platform, and chemical injection was controlled and performed via the Python 3 software (see Methods Software).

Unless specified otherwise, a microfluidic experiment would start with a cleaning and preparation phase, during which all chemicals were injected simultaneously to ensure that no air bubbles remained in the tubing. The duration of this phase was based on the maximum length between any injection inlet in the platform and the flow cell, giving sufficient time for all chemicals to completely go through the platform. Pure PBS was then injected using the Sol pump to collect the light intensity reference for the absorbance measurement. The chemicals providing a baseline for the experiment were then injected at a constant flow rate for a sufficient time to remove any signal from the setup. Next, chemical signals were generated and transmitted by injecting Y and/or P, and data collection was initiated. The injection pattern of Y and P was specific to each experiment and followed the flow rate profiles described in the above methods. After transmitting all the signals from the transmitter, the chemicals providing the baseline were injected again until all the signals passed the flow cell.

The specific experimental settings and injection patterns performed for all the experiments presented in all the figures are provided and thoroughly detailed in Supplementary Note 6.

### Absorbance and pH quantification

Intensity data were collected by the UV-Vis spectrometer with a sampling rate of 0.2 Hz and an integration time of 500 ms. As mentioned in the above methods, the intensity baseline required for the computation of the absorbance was recorded prior to the experiment by measuring a pure PBS solution for 30 s and averaging the intensity over time.

Experimental data were analysed in Python to calculate the absorbance for each wavelength $\lambda$ using the equation

$$A(\lambda) = \log_{10}\left[\frac{I_0(\lambda)}{I(\lambda)}\right], \qquad (9)$$

where $I(\lambda)$ and $I_0(\lambda)$ represent the intensity of the solution being analysed and the intensity baseline at wavelength $\lambda$, respectively. The revised pH of the solution, noted $pH_r$, was directly computed from the absorbance at 453 and 616 nm, corresponding to the wavelength of maximum absorbance for HBTB and $BTB^-$ respectively, using the equation

$$pH_r = \log_{10}\left\{\frac{\left(\frac{A(616)}{A(453)}\right)(\epsilon_A(453) - \epsilon_B(453)) - (\epsilon_A(616) - \epsilon_B(616))}{K_C\epsilon_B(616) - K_C\left(\frac{A(616)}{A(453)}\right)\epsilon_B(453)} - \frac{1}{K_C}\right\}. \qquad (10)$$

where $K_C = 7.9 \times 10^{-8}\ mol\ L^{-1}$, $\epsilon_A(453) = 7370\ L\ mol^{-1}$, $\epsilon_A(616) = 1290\ L\ mol^{-1}$, $\epsilon_B(453) = 240\ L\ mol^{-1}$, and $\epsilon_B(616) = 31{,}800\ L\ mol^{-1}$. The development of the above mathematical model was provided in Supplementary Note 2.

### Pulse analysis and bit decoding

Pulses are defined as signals above the pH baseline, measured during a period significantly long enough to be not considered as background noise, and characterised by low (off) and high (on) pH levels. The pulses were detected from the time-varying pH data using Python and analysed to measure the observation time $T_o$, the pulse width $T_w$ and the off-on transition time $\Delta T$ defined in Supplementary Note 1. The detection threshold was defined using the measured average values of the low and high pH levels of the signal and set as $-3$ dB. The detection threshold was then directly used in a basic step detection algorithm,

provided in Supplementary Note 5, to localise in time all the changes between $\mu_{\text{off}}$ and $\mu_{\text{on}}$, called time steps $t_0$, thus localising all the pulses in the signal. For each detected pulse, $T_o$ and $T_w$ were directly calculated from the localised step times. The $\Delta T$ of each edge of a pulse was extracted by fitting the edge to a sigmoid function

$$\text{pH} = \text{pH}_{\text{on}} + \frac{\text{pH}_{\text{off}} - \text{pH}_{\text{on}}}{1 + \exp\left(\frac{t - t_0}{\text{dt}}\right)}, \tag{11}$$

with $t_0$ the localised time step, and $\text{pH}_{\text{off}}$ and $\text{pH}_{\text{on}}$ the fitted values for respectively the low and high pH levels. $\Delta T$ was directly computed from the transition rate dt of the sigmoid such as $\Delta T = (\text{pH}_{\text{on}} - \text{pH}_{\text{off}})/(4\text{dt})$.

Bit sequences transmitted with the MIMIC platform were decoded from the time-varying pH signal by binning that signal in time with bins of duration $T_b$. For each bin, bit-1 was identified if the pH went above a detection threshold, or identified as bit-0 otherwise. The bit interval $T_b$ and the bit detection threshold were computed through the analysis of two consecutive bit-1 emitted before each message, using the pulse analysis method described above, with $T_b$ corresponding to the difference between the observation time $T_o$ of the two pulses, and the bit detection threshold corresponding to the pulse detection threshold of these pulses. The decoded bits were then grouped by 7 and translated into ASCII characters.

## Communication reliability

The communication reliability was quantified by BER, which is defined as

$$\text{BER} = \frac{\text{The number of bit errors}}{\text{The total number of bits}}. \tag{12}$$

To ensure that all measurements of the BER in different conditions can be accurately compared, a 100-bit sequence was randomly generated once (including 53 bit-0 and 47 bit-1 and presented in Supplementary note 6.8) and was used for each individual measurement.

## Statistics reproducibility

The reproducibility of the chemical reactions presented in equations (1), (3), and (4) as well as their responses in pH and absorbance was assessed by replacing all solutions with fresh ones every two weeks at the latest, by measuring their pH and absorbance, and by comparing these values with the values of previous measurements. Solutions were rejected and re-prepared if the measurements did not match. All data collected from a validated solution were then used in this work without any further rejection.

Each point presented on all the graphs corresponds to a single measurement. A minimum of 10 data points were collected for use with a non-linear model (either to fit experimental data or to compare experimental data with a theoretical prediction), while a minimum of 5 data points were collected for use with a linear model.

With the exception of Fig. 6c, all the pH signals over time presented in the main text were the only signals measured for these experiments. The signals presented in Fig. 6c serve strictly as an illustration of the specific phenomenons observed in Fig. 6d, and these 5 signals shown in details were selected over the 14 signals collected and used in Fig. 6d to cover a large range of $T_b$ values above and below $T_b^*$.

## Reporting summary

Further information on research design is available in the Nature Portfolio Reporting Summary linked to this article.

## Data availability

The data that support the findings of this study are available from Zenodo[56] and from the corresponding author upon request.

## Code availability

Files to 3D-print the syringe pumps, the Arduino codes for the pumps, and Python codes for the control software of the MIMIC platform are available at https://github.com/kcl-yansha/MolCommUI. The codes supporting the data analysis are available from the corresponding author upon request.

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

## Acknowledgements

We thank Dr. Mingfeng Wang at Brunel University London for fruitful conversations and the precious help with setting the PID controller to regulate the flow rate of the pump. We thank the Department of Physics at King's College London for allowing us to conduct in their premises all the experiments used to collect the data for this paper, as well as the Department of Surgery and Cancer at Imperial College London for allowing us to calibrate our equipment in their premises. This work was supported by an EPSRC/UKRI New Investigator Award (EP/T000937/1) to Y.D. and an EPSRC/UKRI Innovation Fellowship (EP/S001603/1) to A.S-R.

## Author contributions

Y.D. conceived of the project. All the authors contributed to the microfluidic design of the platform. D.B. and Y.D. developed the theoretical framework of the platform. V.W. and A.S-R. designed and 3D printed the parts of the microfluidic prototype. A.S-R. and V.W. developed several chemical reaction systems for potential use on the platform. V.W. designed the chemical reaction system reported here. V.W. built the MIMIC platform with support from A.S-R.. D.B. and V.W. wrote the Arduino code for the syringe pump, and V.W. wrote the Python code for the control software. V.W. and D.B. performed all experiments on the platform. V.W. analysed the data, all the authors contributed to data interpretation. V.W. and D.B. prepared drafts of the paper, Y.D. and A.S-R. critically reviewed and edited drafts. Y.D. and A.S-R. supervised the programme of research.

## Competing interests

The authors declare no competing interests.
