## [Peer Review File · Nature Communications]

Real-time signal processing via chemical reactions for a microfluidic molecular communication systemReviewers' Comments:

Reviewer #1:

Remarks to the Author:

Molecular communication is getting out of its niche and is about to turn into a widely adopted concept with applications in industrial environment but, of course, particularly in life sciences and medicine. In this line, the authors investigated a very specific communication channel using concentrations of sodium hydroxide as a transmitter. This can easily be detected due to the changing pH level in the fluid medium.

Conceptually, the paper is following earlier work on molecular communication - most of which rather theoretical based on mathematical or simulation models. Going beyond, for example the authors in reference [18] conducted experimental studies using magnetic particles as information carriers. There have been a couple of follow-up studies combining experimentation with larger scale simulation (not cited here).

Talking about the submitted paper, I have to say that I very much enjoyed reading the manuscript. It is overall very well prepared and easy to follow. I particularly like the systems approach to molecular communication covering the theoretical background as well as a (rather complicated) experimental design.

Both the methodology as well as the scientific approach to gain understanding of the concentration based molecular communication are sounds and effective. The presented results are quite convincing that the approach is very robust and can deliver digital data over the molecular communication channel in an experimental setup.

Having said that, I also see two weaknesses that need to be discussed further:

First, the encoding of information using concentrations of sodium hydroxide leads, as intended, to varying pH levels. This will have a significant impact on the system under study. Assuming application in disease treatments, such changes will have a significant impact on the physiology of the (human) body. The applicability may therefore be quite limited. Considering industrial scenarios, also here the pH level may have significant impacts such as fostering corrosion. This is quite different, for example, to the cited paper using magnetizable particles.

Second, the authors explicitly state that the system supports "high transmission speeds". I understand that such statements are always in relation to the system under study but results indicate symbol times in the order of tens of minutes, thus, message rates in the order of hours. This is not very high speed. This is even significantly slower than the solutions cited in the paper (e.g., reference [18]). There are other techniques currently discussed for delivering messages, e.g., in the human body by means of nanobots (think of biochemical artifacts), which travel within the blood system much than the faster suggested concentration changes.

Reviewer #2:

Remarks to the Author:

In this work, the authors present a liquid-based molecular communication system using chemical reactions to model different signal-processing components. I like the simplicity of the model reactions and the usage of chemical properties to tackle the different aspects of signal handling. However, I see a significant disconnect between the problem/motivations and the methods presented. I also believe that the comparison with existing work is not presented clearly. More specifically, reading through the introduction, existing work is mentioned very superficially; then there seem to be two main motivations presented 1) that the existing methods rely on electrical signals in the system; however,

the presented system also uses electronic components (Arduino, Spectrometer...) which weakens this motivation 2) transmission over a longer distance; however the methods presented does not really explain how the contribution tackles this issues, is it just about using the liquid instead of gas? Better pumps? Correction mechanism? The Waveform Design section? If this is presented as a motivation/achievement, the direct relation between the contribution and the result should be stated clearly. Therefore, while the presented methods are interesting, I do not recommend publishing in the current form since it is not clear what specific problems the contribution is tackling and what is the direct relationship between the "novel" contributions and solving these problems.

Below are some comments about different parts of the paper:

* In general, the introduction is probably the weakest part of the paper and could be much better. For example, the first five lines are used to discuss the invention of the telephone two centuries ago instead of a more concise introduction to the field and the problem.

* "Liquid transceivers are not new in communication engineering, examples include Bell's liquid transmitter.," given that there are many examples, are not there more recent than 1876? Something from the literature? I would prefer to see more comparisons with liquid-based systems in general.

* I prefer to see more specific problem motivation in the intro and more insights about how they are tackled instead of just claiming that the system solves all the problems without giving clear intuition for why it solves them, at least.

* "To address this challenge, we develop a liquid-based microfluidic molecular communication (MIMIC) platform.." as mentioned earlier, this is supposed to be the main contribution, but it reads to me as "We designed a system that solves the existing problem," and that is it, what are the main ideas that made the system solve it? There are more details in the following sections, but I believe some of them should be shown in the introduction, at least an overview, and how they connect to the problems being tackled.

* "A flow chemistry challenge, jointly designing the microfluidic geometry ...transformation for timely communication," while I understood the statement from reading the paper, I think it can be rewritten more clearly.

* The part about using a suppressor "...controlling the timing of an interaction with a signal suppressor P to completely consume Y." is not clear; my understanding is that it separates different signals (e.g., between different bits), but it needs to be explained clearly.

* "To fulfill this requirement, we designed the microfluidic geometry of the platform to control the moment at which the solution would reach the next location.." similar question, how? What did the authors do to tackle this other than that the system solves it?

* This is a minor comment, but the terminology for "amplification" is a bit inaccurate for me since the original NaOH is not detectable using UV-Visible spectroscopy; then the BTB seems more of a detector than amplifier?

* I'm unsure if it is trimmed or missing a label, but I cannot find Fig 3.d? It is mentioned in the text and the caption, but there is no label 3.d.

* "While a pH-meter is a convenient and efficient probe on acid-base solutions, it cannot be applied to other chemical reactions due to its specificity." while it is true that the system using the spectrometer for detection, all the provided reactions seem to rely strongly on pH based computation, are their variants for the system that can work with a different category of reactions to support this statement?

* "So far, the longest communication distance reported in the MC platform literature was 4 m..but also the transmission speed to the limit of our hardware." same remark, what part of the contribution made it possible to go from 4m to 25m?

* Given that there are 3D-printed components for the system, I suggest that the design files get attached to manuscript files if someone wants to replicate them.

* If a pH meter was used in preparing the chemical solutions, I suggest listing it.

* I would prefer to see more numbers/graphs/tables for "4.7 Communication Reliability," giving more details about the error rate; even the Supplementary Information only discusses the measurement method, not the results.

Reviewer #3:

Remarks to the Author:

This paper describes the design and construction of a microfluidic molecular communications platform for transmitting digital signals over distances using varying concentration of sodium hydroxide. Signal shaping, thresholding, amplification, and detection by optimal (UV-Vis) spectrometer are described. Communication distance of 25 meters is achieved and 100 bits were reliably transmitted. The possible applications of molecular communication are particularly exciting in biological applications: "where the transmitter could efficiently communicate with cells and their membranes (e.g., drug delivery, DNA sensing) and the receiver could decode and read chemical signals emitted by an organism (e.g., cell culture monitoring, detection of illness-specific biomarkers)."

Given the particularly exciting applications of molecular communications in the biological context, I had a hard time connecting the system demonstrated to this context. In particular, manipulating pH / sodium hydroxide is not something that can be reasonably be done in the body or in a biologically compatible way. In this regard, communicating with nucleic acids or bio-compatible small molecules seems more aligned with the intended goal. If the intent was more on demonstrating certain fundamental principles, I wasn't sure what general ideas could be more generally extracted from the demonstrated scheme.

The advancement of the state of the art in terms of quantitative measures is limited because of tradeoffs with the transmission bit rate. The authors point out the much longer communication channel and a optical readout method achieved. However, prior work using pH changes has achieved significantly larger transmission bit rate: eg ref [17] achieves 4 bits per second compared with 0.16 bits per minute in figure 4. The authors should clarify what characteristic of their system are incompatible with the faster transmission rate. The authors may also want to clarify what limited the communication distance.

If I am understanding correctly, the amplification was performed at the point of conversion from pH to Absorbance (Fig 2a), which is non-linear and exhibits a response satisfying condition (2) of the signal amplification desiderata ("in the presence of Y, the concentration value of O should not be influenced by the concentration of input Y"). Is this consistent with equation (1) which shows that Y acts catalytically to produce O?

Other than the above major issues, the paper is generally well-written. The figures are very helpful and nice. I found a minor grammar / clarification issues:

- line 073: awkward phrasing: "The likeliness of chemical reactions over air can be constrained drastically..."
- line 128: challenge (i) is very long and hard to understand.
- line 190: I didn't understand the point about compensating the variation of other pumps.

- figure 1 caption: "provided for each equipment" => "provided for each piece of equipment"
- line 322: Fig. 2c?
- line 628: There is no Fig. 5e
- line 635: "will be hard to be decoded" => "will be hard to decode"
- line 637: rephrase: "signal distortion induced by a twice consumption of Y"
- equation (4): more explanation justifying this equation would be helpful

Reviewer #4:

Remarks to the Author:

Summary

The objective and key findings of the study was to demonstrate liquid based digital communication using ASCII characters, over a microfluidic channel for a distance greater than reported hitherto in the literature. The headline capability is reliable transmission of 100 bits over 25m. The digital information is encoded using chemical concentration levels and the information decoded using UV-VIS spectrometry. The system is optimized to maximise the data rate and minimise the bit error rate. The system is therefore novel, and represents a first for this method of communication. It is however very sophisticated (complicated) in achieving a modest communication goal. The paper has 6 figures which are well constructed. There are 30 references which is less than I would expect given the number of papers published in this area.

The study is well designed, the analysis appears correct and sufficient data are presented to enable conclusions to be drawn to validate the headline claims. A limitation of the paper is a discussion on how such a system would be applied in practice and whether there are any advantages over other molecular communication (MC) systems.

The research question is not well articulated. The paper describes what was done (described in both the Abstract and main text) but not why it was done and what is the wider significance (relevance) of the work itself to other workers and of the approach itself. From my perspective this is a major limitation of the paper. There is also some confusion as to the final goal: if it is to demonstrate MC at small dimensions (e.g MC using a microfluidic platform) then what is the purpose of the validation of the technique over 25m?

Title: The title is descriptive, but 'Realizations...' does not read well. I suggest this word is omitted altogether in future revisions.

Abstract: This needs a clear statement of the research question, its significance and relevance to other MC workers and the wider community.

Introduction:

A survey of the literature is given but there are some important omissions which are relevant to the experiments undertaken (e.g. McGuinness et al, 'Experimental and Analytical Analysis of Macro-Scale Molecular Communications within Closed Boundaries', IEEE Trans. on Molecular, Biological and Multi-scale Communications, 5, 1 (2019))

The paper claims (lines 94-97) that 'To achieve the goal ... it is fundamentally important to design a family of pure chemical concentration, signal processing ... blocks that operate in real-time...' The paper however does not say why this is fundamentally important.

Results and Methodology: The figures are clear, well presented and of high quality. The methodology is described in detail and perhaps this section could be shortened. The headline claims of the study of are validated by the information presented. Proof of principle of using liquid chemical concentration and flow chemistry to transmit information is therefore demonstrated. System reproducibility by other

workers is possible with the information provided. However, the system described is complex and very sophisticated for modest information transmission. Unless the authors can demonstrate some key and unique advantages using their approach over other approaches (e.g. using gas concentrations) I believe that adoption by other workers will be very limited.

Discussion: The paper does not clearly explain the implications of the study for the field and its potential future applications. Discussion of the limitations of their technique in comparison with other published MC systems is lacking. The paper claims that the work is a 'major step toward promised MC applications...' (Line 742), but this conclusion is not supported by the paper it would be unlikely in my view, on the basis of what has been reported in the paper that MC using this approach would be adopted.

Minor changes: Apart from the change in title (above) I can see few minor corrections that are needed.

Review conclusion: I do not recommend publication until the major points outlined above are addressed by the authors.

**Response to Reviewers' Comments for
Manuscript Paper ID NCOMMS-23-12790**

Signal Processing via Chemical Reactions for a Microfluidic
Molecular Communication System

Addressed Comments for publication to NATURE COMMUNICATIONS by the authors

August 7, 2023

Dear Reviewers,

We would like to thank the reviewers for reviewing our paper. We are also indebted to the reviewers for their many constructive and helpful comments. We have addressed all the comments and substantially updated the original manuscript. To enhance the legibility of this response letter, all the reviewers' comments are typeset in *italic font* and **blue**, and our responses are written in plain font. Rephrased sentences are typeset in **red**.

Yours sincerely,
The authors

Contents

I Response to Reviewer #1	4
I.1 R1 Comment 1	5
I.2 R1 Comment 2	7
II Response to Reviewer #2	9
II.1 R2 Comment 1	13
II.2 R2 Comment 2	15
II.3 R2 Comment 3	16
II.4 R2 Comment 4	18
II.5 R2 Comment 5	18
II.6 R2 Comment 6	19
II.7 R2 Comment 7	20
II.8 R2 Comment 8	21
II.9 R2 Comment 9	23
II.10 R2 Comment 10	23
II.11 R2 Comment 11	25
II.12 R2 Comment 12	27
II.13 R2 Comment 13	27
II.14 R2 Comment 14	27
III Response to Reviewer #3	30
III.1 R3 Comment 1	32
III.2 R3 Comment 2	34
III.3 R3 Comment 3	37
IV Response to Reviewer #4	40
IV.1 R4 Comment 1	47
IV.2 R4 Comment 2	47
IV.3 R4 Comment 3	49
IV.4 R4 Comment 4	50
IV.5 R4 Comment 5	54
IV.6 R4 Comment 6	57

I Response to Reviewer #1

General comment

Molecular communication is getting out of its niche and is about to turn into a widely adopted concept with applications in industrial environment but, of course, particularly in life sciences and medicine. In this line, the authors investigated a very specific communication channel using concentrations of sodium hydroxide as a transmitter. This can easily be detected due to the changing pH level in the fluid medium.

Conceptually, the paper is following earlier work on molecular communication - most of which rather theoretical based on mathematical or simulation models. Going beyond, for example the authors in reference [18] conducted experimental studies using magnetic particles as information carriers. There have been a couple of follow-up studies combining experimentation with larger scale simulation (not cited here).

Talking about the submitted paper, I have to say that I very much enjoyed reading the manuscript. It is overall very well prepared and easy to follow. I particularly like the systems approach to molecular communication covering the theoretical background as well as a (rather complicated) experimental design.

Both the methodology as well as the scientific approach to gain understanding of the concentration based molecular communication are sounds and effective. The presented results are quite convincing that the approach is very robust and can deliver digital data over the molecular communication channel in an experimental setup.

Having said that, I also see two weaknesses that need to be discussed further.

We would like to thank the reviewer for briefly summarizing the development of MC, recognizing the novelty of our paper, and their valuable suggestions for improving the quality of the manuscript. We also thank the reviewer for pointing out that there have been a couple of studies combining experimentation with larger-scale simulations. In the revised manuscript, we added three related references [18, 30, 31] when we discussed that various molecules have demonstrated their feasibility for information exchange. We have also revised the paper in line with the reviewer's following comments, thereby improving the contributions and the clarity of the paper accordingly.

[18] Koo, B.-H. *et al.* Molecular MIMO: From theory to prototype. *IEEE J. Sel. Areas Commun.* **34**, 600–614 (2016).

[30] Wicke, W. *et al.* Experimental system for molecular communication in pipe flow with magnetic nanoparticles. *IEEE Trans. Mol. Biol. Multi-Scale Commun.* **8**, 56–71 (2022).

[31] Bartunik, M., Fischer, G. & Kirchner, J. The development of a biocompatible testbed for molecular communication with magnetic nanoparticles. *IEEE Trans. Mol. Biol. Multi-Scale Commun.* **9**, 179–190

(2023).

I.1 R1 Comment 1

First, the encoding of information using concentrations of sodium hydroxide leads, as intended, to varying pH levels. This will have a significant impact on the system under study. Assuming application in disease treatments, such changes will have a significant impact on the physiology of the (human) body. The applicability may therefore be quite limited. Considering industrial scenarios, also here the pH level may have significant impacts such as fostering corrosion. This is quite different, for example, to the cited paper using magnetizable particles.

Response:

We thank the reviewer for this constructive comment, demonstrating that we need to edit the manuscript to better explain our objectives and its motivations. In this work, we prove experimentally that all the signal processing required in MC to properly encode and decode signals (e.g., signal shaping, thresholding, amplification) can be performed entirely in the molecular domain instead of relying on external electrical devices. As we wanted our demonstration to be easily reachable to the entire MC community, as well as other communities such as chemistry and biology, we decided to use a set of chemical reactions universally known, thus helping any reader to easily understand and transpose their own ideas and needs to our design presented here. The pH-based reactions are certainly one of the most famous reactions in all domains, and pH variations were even already used in previous MC testbeds to represent the bit messages [22, 23]. Moreover, controlling the pH of different solutions is always critical since acidity can be required to enable chemical reactions or induce undesired effects such as corrosion. However, as the reviewers rightfully pointed out, the direct applications in medicine and industry of pH-based reactions are quite limited and can be perceived as less interesting than other reactions. But the reactions used in our study have been selected to demonstrate and quantify the feasibility of our general chemical design for device-free chemical signal processing functions. The principle of these reactions, i.e., the way they modify the signal to ensure communication, goes beyond the simple pH-based reactions we present, and can be applied to any kind of reactions that would be relevant to MC, medicine, or industry. For instance, in an application using fluorescently labelled DNA, one could directly replace the suppressing reaction ($Y + P$) presented in this paper with a reaction between the labelled DNA and a blackhole fluorescence quencher, where the latter would completely suppress the fluorescence of the former. While each of these specific reactions would demonstrate one direct application in medicine or biology, their specificity would make it harder to transpose the design principle of our work to other applications. This motivated our decision to use in this paper only pH-based reactions, as we believe that the scope of

our paper was to present a versatile method to a large audience, where other (bio)chemical reactions may have drastically reduced our versatility. We do understand and take into account the concerns of the reviewer on the impact of our study, and we have therefore edited the Introduction and Discussion of our paper to better clarify the objective sought using pH-based reactions in our study and prevent any further misunderstanding. We have also added a discussion on potential applications and provided some corresponding potential reactions. Therefore, we revised our manuscript as follows:

The fifth paragraph of the introduction (page 3):

To ensure that the novelties of our platform can be easily understood by different disciplines, we choose a selection of universally known pH-based reactions, based on sodium hydroxide (NaOH) and hydrochloric acid (HCl), to demonstrate and quantify the feasibility of our general chemical reaction design for electronic-free chemical signal processing functions. While these reactions might not lead to a specific application in medicine or industry, their universality can perfectly illustrate that the design principles behind our platform can be applied to any kind of reaction that would be relevant to a specific application. . . . Benefiting from the modularity of our platform, our MC transmitter and receiver are expected to operate individually for microfluidic-based applications by replacing the acid and base with application-specific chemicals and designing the corresponding chemical-reaction-based signal processing functions. . . .

The fifth paragraph of The discussion (page 16):

It is important to highlight the fact that all the experiments and equipment presented in this work are used as a strict proof-of-concept platform to validate the feasibility of chemical signal processing using the universally known acid and base-based chemical reactions as an example. However, the novelty of our MIMIC platform is not limited to these reactions and hardware, and practical applications of our MIMIC platform and its processing functions, e.g., in biology, medicine, or environmental sciences, can be achieved by adapting the chemical reactions and the platform accordingly. The versatility of our design makes it easy to replace the chemicals and the reactions used in this work with corresponding molecules specifically dedicated to the sought application. For instance, with a fluorescent dye as the signal carrier, signal suppression (i.e., used for signal shaping and thresholding) can be achieved with a black hole quencher, while signal amplification and detection can be achieved using another fluorescent probe by Förster resonance energy transfer (FRET) [51, 52]. . . .

I.2 R1 Comment 2

Second, the authors explicitly state that the system supports “high transmission speeds”. I understand that such statements are always in relation to the system under study but results indicate symbol times in the order of tens of minutes, thus, message rates in the order of hours. This is not very high speed. This is even significantly slower than the solutions cited in the paper (e.g., reference [18]). There are other techniques currently discussed for delivering messages, e.g., in the human body by means of nanobots (think of biochemical artifacts), which travel within the blood system much than the faster suggested concentration changes.

Response:

We thank the reviewer for pointing out the clarity issue in our use of the term “high transmission speeds”. We also agree with the reviewer that it may bring confusion to say our platform achieves “high transmission speed”. We intend to describe the speeds relative to the performance of our platform design, e.g., by using “speed x1”, “x2”. In this context, the term “high transmission speeds” designates the highest transmission speeds achievable by our platform and is not meant to compare with other transmission rates reported in the literature. The maximum transmission speed of our platform is limited by the reaction time of the different chemical reactions used to process signals, which constrained our choice of tubing length and pumps. If one wanted to reproduce exactly our platform with the same equipment and design as described in the Methods and provided in our GitHub repo, the highest transmission rates they would achieve would be the one we quantify here as the “highest transmission speeds”. By contrast, other MC platforms mainly focused on validating the feasibility of exchanging information via different particles, and there are no reactions designed for signal processing at their transceivers. Their setup was therefore designed to optimise the transmission rate of molecule propagation, explaining why our transmission speeds would in comparison appear as low with the joint consideration of chemical-reaction-based signal processing and molecule propagation. We do understand how our current use of the term “high transmission speeds” can lead to confusion from the reader. To further clarify this, we revised our manuscript as follows:

The first paragraph of Section 2.6 (page 13):

To explore the limits of the communication performance of our MIMIC platform, we investigated the communication efficiency for transmissions **at the highest speeds achievable by our system and over a long distance.**

The second paragraph of Section 2.6 (page 13):

We further assessed the communication performance by gradually increasing all the injection flow rates by a speed factor of N to their limit governed by the pump hardware, **up to 200 $\mu\text{L}/\text{min}$ for a single pump.** ... These experiments

demonstrate the capacity of our MIMIC platform to achieve high data rates under the constraints set by its design, although the highest possible data rates of our system are found lower than others previously reported in the literature [30, 31].

The fourth paragraph of the discussion (page 19):

... However, while our platform successfully transmitted information at the highest transmission speed achievable by our platform with chemical signal processing capability, this speed is lower than the MC platform without chemical signal processing capability [31, 32]. This is because the highest transmission speed of our platform is limited by the reaction times of all the chemical signal processing functions, as well as the maximum injection speed of our 3D-printed pumps. It is important to note that higher data rates can be achieved by adapting our MIMIC platform with more powerful pumps and other kinds of faster chemical reactions.

II Response to Reviewer #2

General comment

In this work, the authors present a liquid-based molecular communication system using chemical reactions to model different signal-processing components. I like the simplicity of the model reactions and the usage of chemical properties to tackle the different aspects of signal handling. However, I see a significant disconnect between the problem/motivations and the methods presented. I also believe that the comparison with existing work is not presented clearly. More specifically, reading through the introduction, existing work is mentioned very superficially; then there seem to be two main motivations presented 1) that the existing methods rely on electrical signals in the system; however, the presented system also uses electronic components (Arduino, Spectrometer...) which weakens this motivation 2) transmission over a longer distance; however the methods presented does not really explain how the contribution tackles this issues, is it just about using the liquid instead of gas? Better pumps? Correction mechanism? The Waveform Design section? If this is presented as a motivation/achievement, the direct relation between the contribution and the result should be stated clearly. Therefore, while the presented methods are interesting, I do not recommend publishing in the current form since it is not clear what specific problems the contribution is tackling and what is the direct relationship between the “novel” contributions and solving these problems.

We would like to thank the reviewer for summarizing our manuscript, highlighting these major issues, and their other valuable suggestions for improving the quality of the manuscript. We have revised the paper in line with the reviewer’s comments, thereby improving the contributions and the clarity of the paper accordingly.

Disconnect between the problem/motivations and the methods presented

We thank the reviewer for pointing out that the problem/motivation and our methods were not well connected. The objective of this paper is to design and develop *pure chemical concentration* signal processing blocks for MC. This is motivated by the fact that electronic signal processing devices are difficult to operate efficiently and miniaturize to nano/micrometer-scale and can hardly meet the biocompatible requirement of MC-promised applications. The pure chemical concentration signal processing blocks are not studied in existing gas-based or liquid-based MC platforms as they only focused on validating the feasibility of exchanging information via various types of molecules. To address this research gap, we design a set of chemical reactions to perform signal processing functions for chemical signals over the molecular domain. To clarify the problem/motivations and how it is connected with our methods, we revised the manuscript as follows:

The third paragraph of the introduction (page 2):

... **It is important to note that the electrical devices here are integrated**

with these macroscale instruments to perform signal processing functions (e.g., encoding-decoding [22] and detection algorithms [18, 22, 23, 25, 27]) based solely on electrical bit signals, with the aim of ensuring successful information transmission after propagation. Although it is acceptable to use electronic devices for signal processing when focusing on the information exchange based on chemical molecules, the utilization of electronic devices for signal processing functions can hardly meet the biocompatible, non-invasive, and size-miniaturized requirements of most microscale/nanoscale biochemical applications, e.g., tissue engineering, targeted drug delivery, and immune system enhancement [37]. Thus, it is important to shift from electrical signal processing to chemical signal processing and design a family of pure chemical concentration signal processing (e.g., pulse shaping and thresholding) blocks that operate in real-time for signal generation, detection, amplification, etc. Nevertheless, the experimental implementation of these basic signal processing functions over chemical signals, especially at the microscale/nanoscale, has been ignored so far.

The fourth paragraph of the introduction (page 3):

... Considering that chemical reactions are more likely to occur in the liquid phase [45, 46], here we develop a liquid-based microfluidic molecular communication (MIMIC) platform. Specifically, through a joint design of a selection of chemical reactions and of a finely tuned microfluidic geometry for the transceiver, our MC transmitter is capable of shaping the transmitted signals, and the MC receiver is capable of detecting an incoming signal exceeding a pre-defined threshold and amplifying it to a user-defined level. Compared to the previously reported platforms, the signal processing functions in our MIMIC platform are realized over the molecular domain and are free of electronic devices, only using macroscale external devices to interface with our transmitter and receiver for chemical injection and quantitative measurement and to provide complete control for the performance evaluation.

Comparison with existing MC work

We thank the reviewer for this comment. We agree that the difference between existing experimental MC platforms and our MIMIC system was not well clarified. As stated in the response to the reviewer’s previous comment on the disconnect between the problem and the methods, existing experimental MC platforms only focused on validating the feasibility of exchanging information via various types of molecules. They performed signal processing functions *before molecule release* and *after molecule reception* inside electronic devices based on *electrical signal processing* rather than *chemical signal processing*, as shown in Fig. 1. However, the utilization of electronic devices for electrical signal processing func-

tions can hardly meet the biocompatible, non-invasive, and size-miniaturized requirements of some microscale/nanoscale biochemical applications. Compared to existing MC platforms, the novelty of our platform is to realize pure concentration-based chemical signal processing blocks via chemical reactions, these functions are performed *after molecule release* and *before molecule reception*, i.e., in the molecular domain (see Fig. 1). Although we also used electronic devices, they are used to interface with our transmitter and receiver for chemical injection and quantitative measurement and provide complete control for platform performance evaluation, but not for performing signal processing functions, including the signal shaping, signal thresholding, and signal amplification and detection functions. To clearly present the difference with existing MC platforms, we revised the introduction, and please refer to our response to the general comment (page 22) on the disconnect between the problem/motivations and the methods.

Figure 1: An illustration of the comparison of (a) existing MC platforms and (b) our MIMIC platform.

The use of electronic components in our system

We thank the reviewer for pointing out that the reason why we used electronic components in our system was not well clarified. Note that the reason we used computer-controlled syringe pumps and a UV-visible spectrometer is to interact with our MC platform to have extensive control of the chemical injection as well as an accurate reading of the signals at

the receiver output. As stated before, the novelty and objective of our work is to develop a new and novel method to process chemical signals in the molecular domains, while all existing methods rely on electronic devices to achieve signal processing functions as shown in Fig. 1 in the last page.

In our work, we present four main signal processing functions that are all entirely performed in the molecular domain: a signal suppression, used to shape the waveform of emitted signals at the transmitter, a signal thresholding, used to mitigate the effect of the noise and interference brought by the propagation channel, and a signal detection and amplification used to retrieve transmitted signals at a pre-selected intensity. All these signal processing functions have been achieved only through a set of chemical reactions, thus demonstrating our claim that our platform can achieve signal processing functions without relying on external electronic devices. It was however essential to prove that all these chemical reactions were fulfilling their roles and achieved the signal processing intended by design. To prove and validate the signal processing capabilities of our designed chemical reactions, it is essential to fully and precisely control our platform using electronic components, which includes when each chemical was injected and when the signal was processed by the designed chemical reactions. This is why we are still using electronic components in our platform, while we can confidently claim that all the signal processing functions are now entirely free of electronic components. To clarify our claim and avoid any confusion on the motivation of our work, we have edited the text of the manuscript as follows:

The fourth paragraph of the introduction (page 3):

... Compared to the previously reported platforms, the signal processing functions in our MIMIC platform are realized over the molecular domain and are free of electronic devices, only using macroscale external devices to interface with our transmitter and receiver for chemical injection and quantitative measurement and to provide complete control for the performance evaluation.

Transmission over a longer distance

It seems that the reviewer misunderstood the motivation of our work here, as we never intend to claim that transmission over a long distance was a motivation or an objective of our paper. During the development of our MIMIC platform, we decided to explore the limits of the communication performance of our platform, which included tests at high speeds and over long distances. Increasing the transmission distance will result in both an increase in the back pressure, making it harder for the pumps of the transmitter to operate, and an increase in the noise and interference. We are pleased to report that our platform design could support communication over distances as long as 25 m and decided to list it as an achievement as well as a result since the previously reported

longest communication distance was 4 m. We believe that this achievement is a result not only due to the careful design of our injection system but also due to the design of chemical-reaction-based signal processing functions. The demonstration of the efficiency of our designed signal processing functions is indeed one of the main motivations and objectives of our paper, and the long-distance communication performance supports our results. To further clarify that the long-distance communication was not a motivation for our work but instead an achievement of our design, we revised our manuscript as follows:

The fifth paragraph of the introduction (page 3):

... Using these reactions, we experimentally demonstrate that our designed MIMIC platform can reliably transmit bit signals with a low bit error rate (BER), at the highest transmission speeds achievable by our hardware and over extremely long distances, testing tubing up to 25 m long. ...

The second paragraph of Section 2.6 (page 13):

This tubing length conveniently corresponds to more than 10 times the average longest straight distance in the human body (head to toe).

The fourth paragraph of the discussion (page 19):

We evaluated the communication performance, and we explored the limits of our platform in communication, specifically to transmit and recover a text message over a long distance and at the highest speeds achievable by our hardware, which imposes higher noise and interference levels. In such conditions, our MIMIC platform was able to maintain a satisfying reliability, even with a distance as long as 25 m. ...

II.1 R2 Comment 1

In general, the introduction is probably the weakest part of the paper and could be much better. For example, the first five lines are used to discuss the invention of the telephone two centuries ago instead of a more concise introduction to the field and the problem.

Response:

We thank the reviewer for pointing out that the introduction could be largely improved. In the revised manuscript, we edited the whole introduction which consists of five paragraphs now. Specifically, we rephrased the discussion of the invention of the telephone and provided a concise introduction to MC and its general development in the first paragraph. We discussed the experimental MC in the second paragraph and introduced the problem that the utilization of electronic devices for signal processing functions can hardly meet the biocompatible, non-invasive, and size-miniaturized requirements of some microscale/nanoscale biochemical applications in the third paragraph. We also stated our

motivation for developing a new and novel method to process chemical concentration signals over the molecular domain in the same paragraph. The fourth paragraph presents how we use chemical reactions to achieve the signal processing functions for chemical signals in the molecular domain, and the fifth paragraph provides an overview of the achievements, advantages, and applications of our platform. The revised first three paragraphs are provided in the following:

The first paragraph of the introduction (page 1):

Communication technologies for transmitting information over distances have been revolutionizing the way humans, sensors, and robots communicate. Since the invention of the telephone by Bell in 1876, significant progress has been made over a century in both wired and wireless communications, with major innovations including transmitting optical signals over optical fibre in a confined cable and electromagnetic wave signals over the air in an unbounded environment. Different from wired and wireless communications that are designed for typical human-to-human, human-to-machine, or machine-to-machine macroscale applications (e.g., mobile communications), molecular communication (MC) was proposed for the first time to explore the exchange of information using chemical molecules, mimicking the communication processes found in nature, such as the transmission of pheromones in the air among individuals and the hormone signaling in fluid media inside the human body [1, 2]. MC research holds immense potential in emerging applications where traditional wired or wireless communications would be unsafe or infeasible, such as intrabody biosensing [3], smart drug delivery [4, 5], and explosive gas monitoring [6].

The second paragraph of the introduction (page 2):

Over the past decade, the main focus of MC research was to theoretically characterize and analyse an MC system using tools and mechanisms from communication engineering, and advancement includes modeling the biophysics of transceiver and propagation [7-9], quantifying communication capacity [10, 11], evaluating and optimizing communication performance [12-14], etc. Aligning with the ongoing theoretical research efforts, researchers also realized the importance of developing experimental MC platforms in validating the proposed theoretical models and facilitating practical MC-based applications [15, 16]. However, the development of an experimental MC platform is complex as it requires multidisciplinary expertise (e.g., communication engineering, chemistry, biology, and physics) [2], and to date only a few gas and liquid-based MC platforms have been successfully prototyped [17-36].

The third paragraph of the introduction (page 2):

Like any communication system, an MC platform can be generally divided into three fundamental parts: a transmitter, a propagation channel, and a receiver. So far, the existing experimental MC platforms focused on the propagation through bounded and unbounded spaces, and their main objective was to demonstrate the feasibility of exchanging information via various chemicals, which includes not only the molecules found in nature, e.g., volatile organic compounds [17-21], acids [22, 23], protons [24], glucose [25], DNA molecules [26], and sodium chloride [27], but also the carriers designed in the lab, e.g., droplets and bubbles [28], artificial magnetic nanoparticles [29-31], fluorophore [32, 33], color pigments [34], and carbon quantum dots [35, 36]. In these platforms, their MC transceivers are interfaced with macroscale instruments (spray [17, 18, 33], pump [20, 23, 25, 27-29, 31, 34-36], pH sensor [22-24], light detector [32, 35, 36]) for the conversion between electrical bit signals and chemical signals, i.e., release and observe information molecules. It is important to note that the electrical devices here are integrated with these macroscale instruments to perform signal processing functions (e.g., encoding-decoding [22] and detection algorithms [18, 22, 23, 25, 27]) based solely on electrical bit signals, with the aim of ensuring successful information transmission after propagation. Although it is acceptable to use electronic devices for signal processing when focusing on the information exchange based on chemical molecules, the utilization of electronic devices for signal processing functions can hardly meet the biocompatible, non-invasive, and size-miniaturized requirements of most microscale/nanoscale biochemical applications, e.g., tissue engineering, targeted drug delivery, and immune system enhancement [37]. Thus, it is important to shift from electrical signal processing to chemical signal processing and design a family of pure chemical concentration signal processing (e.g., pulse shaping and thresholding) blocks that operate in real-time for signal generation, detection, amplification, etc. Nevertheless, the experimental implementation of these basic signal processing functions over chemical signals, especially at the microscale/nanoscale, has been ignored so far.

II.2 R2 Comment 2

“Liquid transceivers are not new in communication engineering, examples include Bell’s liquid transmitter..,” given that there are many examples, are not there more recent than 1876? Something from the literature? I would prefer to see more comparisons with liquid-based systems in general.

Response:

We thank the reviewer for this comment. The reviewer is correct that there are many

examples of liquid-based systems more recent than 1876, such as underwater wireless communications [R1, R2], liquid antennas [R3], and liquid-based MC systems [22-27, 29-32, 34-36].

- Compared to underwater wireless communications and liquid antennas which encode information to optical waves [R1], acoustic waves [R2], and radio-frequency (RF) waves [R3] instead of molecules, it is clear that our system uses molecules as the information carrier. Therefore, underwater wireless communications and liquid antennas are not MC systems unlike our system, and these liquid-based communication systems are designed for totally different communication scenarios and applications.
- Compared to existing liquid-based MC systems [22-27, 29-32, 34-36], the major difference is *how* and *where* signal processing functions are performed. As the focus of existing liquid-based MC systems is to demonstrate the feasibility of exchanging information via different chemical molecules through a propagation channel, signal processing functions were performed either before converting digital bits to chemical signals or after converting chemical signals back to digital bits *inside electronic devices*. By contrast, our objective is to develop a new and novel method to process chemical concentration signals in the molecular domain rather than to process electrical signals in the electric domain, which would bring bio-compatibility and non-invasivity to MC systems and unleash their potential for bio-compatible applications. In particular, we designed three chemical reactions to achieve signal shaping, signal thresholding, and signal amplification and detection functions. Thus, the design principle of our system is totally different from existing liquid-based MC systems.

In the revised manuscript, we have deleted this sentence, but we have added more references to liquid-based MC platforms and provided a clear comparison between existing liquid-based MC systems and our system. Please refer to our response to **Comment 1**.

[R1] Zeng, Z., Fu, S., Zhang, H., Dong, Y. & Cheng, J. A survey of underwater optical wireless communications. *IEEE Commun. Surv. Tutor.* **19**, 204–238 (2017).

[R2] Jiang, S. On securing underwater acoustic networks: A survey. *IEEE Commun. Surv. Tutor.* **21**, 729–752 (2018).

[R3] Wong, K. K., Tong, K.-F., Shen, Y., Chen, Y. & Zhang, Y. Bruce lee-inspired fluid antenna system: Six research topics and the potentials for 6G. *Front. Commun. Netw.* **5** (2022)

II.3 R2 Comment 3

I prefer to see more specific problem motivation in the intro and more insights about how they are tackled instead of just claiming that the system solves all the problems without

giving clear intuition for why it solves them, at least.

Response:

We thank the reviewer for pointing out the motivation issue again and the lack of insights about how we tackled the problem. We respond to these two points separately in the following.

The motivation of the paper

The objective of this paper is to design and develop *pure chemical concentration* signal processing blocks for MC. This is motivated by the fact that existing experimental MC platforms performed signal processing functions using external electronic devices, which can hardly meet the biocompatible and non-invasive requirements of MC-promised applications. The pure chemical concentration signal processing blocks are not studied in existing gas-based or liquid-based MC platforms as they only focused on validating the feasibility of exchanging information via various types of molecules. To provide a clear motivation for performing signal processing functions in the molecular domain, we have rewritten the introduction. Please refer to our response to the general comment on page 9.

How we tackled the problem

To tackle the problem of performing time-varying chemical signal processing over the molecular domain, we designed and implemented a group of chemical reactions with corresponding microfluidic geometry, which is inspired by cells that rely on a complex molecular network to process biochemical signals via reacting molecules [38, 39]. In particular, we designed three chemical reactions to achieve the signal shaping, signal thresholding, and signal amplification and detection functions. The signal shaping function is capable of controlling the width of an emitted signal at the transmitter, while the signal thresholding, amplification, and detection functions can mitigate the noise and interference brought by the propagation channel and visualise transmitted signals at the receiver at a predefined intensity. We also applied these three reactions in different regions of a microfluidic system and finely tuned the microfluidic geometry to ensure that the designed signal processing functions are performed appropriately. To make our MIMIC platform easily reachable by the entire MC community as well as other communities such as chemistry and biology, we used the universally known pH-based reactions and encoded information into the pH of a solution. To further explain how our MIMIC platform solved the signal processing problem, we revised our introduction as follows:

The fourth paragraph of the introduction (page 3):

... Considering that chemical reactions are more likely to occur in the liquid phase [45, 46], **here we develop a liquid-based microfluidic molecular communication (MIMIC) platform. Specifically,** through a joint design of a selec-

tion of chemical reactions and **of a finely tuned microfluidic geometry for the transceiver**, our MC transmitter is capable of shaping the transmitted signals, and the MC receiver is capable of detecting an incoming signal exceeding a predefined threshold and amplifying it to a user-defined level. ...

The fifth paragraph of the introduction (page 3):

To ensure that the novelties of our platform can be easily understood by different disciplines, we choose a selection of universally known pH-based reactions, based on sodium hydroxide (NaOH) and hydrochloric acid (HCl), to demonstrate and quantify the feasibility of our general chemical reaction design for electronic-free chemical signal processing functions. ...

II.4 R2 Comment 4

“To address this challenge, we develop a liquid-based microfluidic molecular communication (MIMIC) platform..” as mentioned earlier, this is supposed to be the main contribution, but it reads to me as “We designed a system that solves the existing problem,” and that is it, what are the main ideas that made the system solve it? There are more details in the following sections, but I believe some of them should be shown in the introduction, at least an overview, and how they connect to the problems being tackled.

Response:

We thank the reviewer for pointing out that it was unclear how we achieved signal processing function over the molecular domain in the introduction. We also agree with the reviewer that this should be provided in the introduction to give readers an overview of our methods. As stated in the response to **Comment 3**, we designed three reactions and applied and regulated them in different regions of a microfluidic system to achieve signal shaping, thresholding, amplification, and detection functions. To make this clear, we revised the introduction of the manuscript, and the corresponding revised sentences are also provided in the following. Please refer to **Comment 3** for more details.

II.5 R2 Comment 5

“A flow chemistry challenge, jointly designing the microfluidic geometry ...transformation for timely communication,” while I understood the statement from reading the paper, I think it can be rewritten more clearly.

Response:

We thank the reviewer for this comment. To clarify the first challenge, we rephrased this challenge in the revised manuscript as:

The first paragraph of Section 2.1 (page 4):

In this study, two critical challenges need to be addressed for the development of a liquid-based MC platform that relies on chemical reactions for signal processing: (i) a flow chemistry challenge, **jointly designing the microfluidic geometry and chemical reactions to regulate the occurrence of reactions, which enables the process of time-varying concentration signals in the molecular domain and timely communication**

II.6 R2 Comment 6

The part about using a suppressor “..controlling the timing of an interaction with a signal suppressor P to completely consume Y.” is not clear; my understanding is that it separates different signals (e.g., between different bits), but it needs to be explained clearly.

Response:

We thank the reviewer for pointing out this unclarity in our description of the experiment. The part quoted by the reviewer does not discuss the signal separation, but the characteristics of the reaction between the signal Y and the signal suppressor P. While this reaction can be used to separate different signals, the part quoted by the reviewer only describes the role of P as entirely consuming Y, and that this reaction can be delayed in time using the microfluidic geometry to allow the emission of Y at the outlet of the transmitter during a given time interval. We believe the reviewer is confusing here these two different parts of our work related to the signal suppression. To clarify, this is because the signal suppression reaction $Y + P$ is discussed twice: (i) the $Y + P$ reaction itself, where P completely consumes Y, mentioned in Section 2.1 and thoroughly discussed in Section 2.4, and (ii) the application of this reaction to the signal processing and bit transmission, where this reaction separates different bits, discussed in Sections 2.7 and 2.8.

For a better understanding of the confusion, we will discuss here the text quoted by the reviewer thoroughly. In our design, the signal Y and the signal suppressor P are injected during *the same time interval*. Since P is designed to completely consume all Y, a synchronised injection would result in a complete suppression of Y, therefore no signal would be emitted by the transmitter. In theory, we could just delay the injection of P, and this delay between the injection of Y and P would be the width of the resulting square-like pulse at the transmitter outlet, as illustrated in Fig. 3C and demonstrated in Fig. 5A. In order to achieve signal processing functions by the flow chemistry setup rather than external electronic devices, we also investigated how the microfluidic geometry of the platform can be modified to delay the propagation of P to the chemical reactor instead of delaying the injection time, which is demonstrated in Fig. 5B. To introduce both methods within the same sentence, we discussed in our manuscript how the “timing” of the reaction is controlled. To make this distinction between the $Y + P$ reaction and its impact on the signal more clear, we revised the manuscript as follows:

The fourth paragraph of Section 2.1 (page 4):

With the injection of signal Y at the transmitter inlet during a given time interval, we can adjust the duration of the signal Y emitted at the transmitter outlet by injecting a signal suppressor P within the same time interval. The signal suppressor P completely consumes Y and the duration of emitted signal Y is controlled by the timing of the interaction between Y and P, either by delaying the injection of P or its propagation.

II.7 R2 Comment 7

“To fulfill this requirement, we designed the microfluidic geometry of the platform to control the moment at which the solution would reach the next location.” similar question, how? What did the authors do to tackle this other than that the system solves it?

Response:

We thank the reviewer for this comment. We understand that our initial presentation of the importance of a careful design of the microfluidic geometry lacked clarity. We responded to the reviewer’s two questions separately in the following.

How to control when a solution would reach the next location?

Without adjusting the flow rate, we can control the propagation time of a solution between two locations by adjusting the tubing length between these locations. For a given flow rate Q , a solution will propagate through a tubing with the inner cross-section area S at a velocity Q/S . Given a tubing length of L , the time it takes for the solution to propagate from one end to the other is calculated by $T = LS/Q$. Therefore, by changing the length L of the tubing we can directly set at which time T the solution will reach the end of the tubing.

This is specifically important to ensure that each reaction is complete before the solution of interest reaches the next critical location. Since the critical parameter here is the reaction time T_R , we designed the minimum tubing length that allows for the completion of a reaction as $L_{\min} = QT_R/S$. Therefore, the selected length of tubing needs to satisfy $L \geq L_{\min}$. We can note here that the value of the reaction time T_R can be directly measured in the microfluidic setup, and we provided the details of measuring T_R for reaction $Y + \text{Amp} \longrightarrow Y + \text{O}$ as an example in Supplementary Information 3. We also used this careful tubing length tuning to adjust the propagation time of the solution P thus the shape of the signal at the transmitter outlet, as thoroughly discussed in our response to **Comment 6** on page 19.

What did the authors do to tackle this other than that the system solves it?

We tackled this by carefully designing and fine-tuning the lengths of different tubings to ensure the completion of all the signal processing chemical reactions. Prior work on MC

experimental systems only focused on the propagation of information molecules from one location to another, and no chemical reactions were involved. Prior reported microfluidic platforms usually only consist of a single tubing connecting two locations, and the tubing length was only limited by the distance between these two points. By introducing chemical reactions to process molecular signals as the main concept behind our MIMIC platform, we investigated the minimum constraint on tubing length, thus the whole microfluidic geometry of the platform, to control the reactions and ensure that the signals are processed as intended. To clarify how the tubing length is selected and how essential it is in the design of our MIMIC platform, we revised the manuscript as follows:

The fifth paragraph of Section 2.1 (page 5):

Using chemical reactions in microfluidic setups to process the signal imposes a critical implementation requirement: each chemical reaction should be completed before the solution reaches the next important location. **In a microfluidic setup, it is possible to control the time it takes for a solution to go from one point to another by adjusting the tubing length separating these two points. Given a tubing of length L and inner cross-section area S , the time T it takes for the solution to propagate is calculated as $T = LS/Q$, where Q is the flow rate of the solution. Therefore, by changing the length L of the tubing we can directly set at which time T the solution will reach the end of the tubing. To ensure that a chemical reaction is complete before the solution reaches the end of the tubing length, the corresponding minimum tubing length L_{\min} that will allow for the completion of the reaction should be calculated and used as a condition on the tubing length in the design of our MIMIC platform. In particular, L_{\min} can be calculated for any chemical reaction via $L_{\min} = QT_{\text{R}}/S$, where T_{R} is the reaction time of that reaction. Note that the value of the reaction time T_{R} can be directly measured in the microfluidic setup by measuring a quantity related to the reaction yield (see Supplementary Information 3).**

II.8 R2 Comment 8

This is a minor comment, but the terminology for “amplification” is a bit inaccurate for me since the original NaOH is not detectable using UV-Visible spectroscopy; then the BTB seems more of a detector than amplifier?

Response:

We thank the reviewer for this comment. We do believe that the terminology for “amplification” is accurate in the context of our work, as BTB plays a key amplifying role in the communication process, even if it also serves as a detector. This ambivalence is already highlighted in the title of section 2.2 “Signal amplification and detection”.

While the detection role of BTB is rather straightforward to see (making invisible NaOH visible to the UV-Visible spectrometer) its amplification function is also thoroughly studied in the paper due to its significance, specifically in section 2.2. As illustrated in Fig. 2C-(iii), the intensity of the output signal O (i.e., BTB in its base form) can be indeed tuned by changing the concentration of BTB at the input of the receiver. We have also demonstrated that the intensity of the output signal does not depend on the concentration of Y in the presence of Y. Both of these points prove the amplification property of BTB as used in our platform.

While the ambivalence in the nature of BTB could allow us to use either amplifier or detector to refer to the nature of its function, for clarity in the text we had to select only one term. Our decision to use “amplifier” instead of “detector” is supported by multiple arguments. First, this term was already used in previous theoretical studies for this reaction [42, 43]. Moreover, the need for a detection reaction is limited to situations where the incoming signal cannot be detected while the amplification reaction, essential to select the intensity of the signal at the receiver output, could potentially be applied to all chemicals. In order to clarify further that BTB serves both roles in our work, but brings more significance to its amplifier function, we modified the text in our revised manuscript:

The first paragraph of Section 2.2 (page 7):

In order to achieve signal amplification at the receiver of our platform just before the signal is converted by the UV-Vis spectrometer into an electrical signal, our amplification reaction at the receiver should follow two design principles: 1) in the presence of Y, the concentration value of O should be a constant value determined by the user and not influenced by the concentration of input Y, while 2) the produced output signal O should be visible and quantifiable to the spectrometer. It should also follow the implementation requirement that the concentration of O should only be determined by this reaction rather than influenced by the hydrodynamics of the testbed. Following these two design principles and fulfilling this requirement will make the chemical Amp the signal amplifier in our design. Moreover, since in our MIMIC platform the signal Y, with the NaOH solution as the main component, is transparent and cannot be detected at the receiver using UV-Visible spectroscopy, the chemical Amp acts as both the signal amplifier and signal detector at the same time. Despite having both properties, we will only be referring in text to Amp as the signal amplifier.

The third paragraph of Section 2.2 (page 7):

Through the catalytic reaction of HBTB with NaOH, the evolution of the concentration of output BTB⁻ with the input concentration of NaOH is non-

linear and can be mathematically described by

$$[\text{BTB}^-] = [\text{BTB}]_{\text{Tot}} \frac{10^{14} K_C [\text{NaOH}]}{1 + 10^{14} K_C [\text{NaOH}]}, \quad (2)$$

with $[\text{BTB}]_{\text{Tot}}$ as the total concentration of BTB, in both acid and base forms, and K_C as the equilibrium constant of reaction (1) (see Supplementary Information 2.1). This sigmoid evolution implies that outside a narrow transition range, a large variation in $[\text{NaOH}]$ will only result in a small variation of $[\text{BTB}^-]$, resulting in the concentration evolution satisfying the second design principle of the amplification reaction. The conversion of the concentration of BTB^- into the absorbance of the BTB solution is then described by the Beer-Lambert law (Supplementary equation S1). By combining the non-linear conversion of the concentration of NaOH into the concentration of BTB^- with the linear conversion of the concentration of BTB^- into the absorbance of the BTB solution, we obtain a non-linear conversion of NaOH into the BTB absorbance. Through our proposed mathematical transformation of the BTB absorbance into a revised pH value (pH_r), the sigmoid evolution of $[\text{BTB}^-]$ results in a pH_r evolution that spans between a low and a high pH limits. In this way, the actual pH values (pH_a), proportional to $[\text{NaOH}]$, measured outside these limits are converted to either the low or the high revised pH limits (Fig. 2A).

II.9 R2 Comment 9

I'm unsure if it is trimmed or missing a label, but I cannot find Fig 3.d? It is mentioned in the text and the caption, but there is no label 3.d.

Response:

We thank the reviewer for spotting this mistake. In the revised manuscript, we replaced the reference to the nonexistent Fig. 3d with the reference to Fig. 3C to correct this mistake.

II.10 R2 Comment 10

“While a pH-meter is a convenient and efficient probe on acid-base solutions, it cannot be applied to other chemical reactions due to its specificity.” while it is true that the system using the spectrometer for detection, all the provided reactions seem to rely strongly on pH based computation, are their variants for the system that can work with a different category of reactions to support this statement?

Response:

We thank the reviewer for this helpful comment. The design of our MIMIC platform is intended as a demonstration of a plug-and-play adaptive testbed, which can indeed be modified to work with different categories of reactions. One example is fluorescence-based chemicals and reactions, for which the interest in MC applications has been demonstrated in prior work [32]. In this example, a fluorescent dye, to be determined by the final application sought, could act as the signal carrier Y, a corresponding black hole quencher could then serve the role of the signal suppressor P and/or the signal thresholder ThL, and the amplification could be achieved by FRET with another fluorescent dye. With this fluorescence-based design, the use of a pH-meter would be invalid while a spectrometer would be the perfect detector.

The purpose of our platform and of our work is to focus on the presentation of an experimental proof-of-concept design where all the signal processing functions are performed via chemical reactions. In this scope, the priority of the selection of the chemicals and chemical reactions is to make a clear presentation of the principle and kinetics of the different reactions, as well as their efficiency to process concentration signals, while ensuring that the reactions can be easily understandable by a wide audience from multiple disciplines (e.g., chemistry, medicine, biology). This motivated our selection of acids and bases, not only for their simplicity but also because they are widely well-known and used in all disciplines. As correctly observed by the reviewer, a pH-meter would have been a perfect fit for acid-based reactions. However, we decided to use a UV-Visible spectrometer specifically because it is easily accessible to all disciplines and versatile. By contrast, a pH-meter is only specific to acid-based reactions. To clarify the selection of the UV-Visible spectrometer, the acid-based reactions, and the potential applications of our platform with concrete examples of other variants, we have revised the manuscript as follows:

The third paragraph of The discussion (page 16):

The advancement and versatility of our platform are also reflected in the selection of the hardware. The previously published studies on acid and base encoding in MC all used a pH-meter to extract information from concentration variations, while our MIMIC platform used a UV-Visible spectrometer. This choice was made to better illustrate the versatility of our design: while a pH-meter is a convenient solution to quantify the properties of acid-base solutions, it cannot be easily applied to other chemical reactions due to its specificity. By contrast, while a UV-Visible spectrometer is not capable of directly detecting changes in acidity, we have successfully demonstrated how chemical reactions can convert an invisible chemical signal into a visible chemical signal. The capability of our chemical-reaction-based signal processing functions to change the nature of the signal carrier brings a versatility that allows us to adapt the signal to the detector type at the receiver, and combined

with the natural versatility of the UV-Vis spectrometer proves the versatility of our MIMIC platform.

The fifth paragraph of The discussion (page 16):

It is important to highlight the fact that all the experiments and equipment presented in this work are used as a strict proof-of-concept platform to validate the feasibility of chemical signal processing using the universally known acid and base-based chemical reactions as an example. However, the novelty of our MIMIC platform is not limited to these reactions and hardware, and practical applications of our MIMIC platform and its processing functions, e.g., in biology, medicine, or environmental sciences, can be achieved by adapting the chemical reactions and the platform accordingly. The versatility of our design makes it easy to replace the chemicals and the reactions used in this work with corresponding molecules specifically dedicated to the sought application. For instance, with a fluorescent dye as the signal carrier, signal suppression (i.e., used for signal shaping and thresholding) can be achieved with a black hole quencher, while signal amplification and detection can be achieved using another fluorescent probe by Förster resonance energy transfer (FRET) [51, 52]. ...

II.11 R2 Comment 11

“So far, the longest communication distance reported in the MC platform literature was 4 m..but also the transmission speed to the limit of our hardware.” same remark, what part of the contribution made it possible to go from 4m to 25m?

Response:

We thank the reviewer for this comment which highlights a lack of clarity in our conclusions. The distance of 4 m reported in the literature has been nowhere defined as a limit for other platforms, we simply added it in the discussion to acknowledge the previous achievement of other MC platforms. We are not comparing the achievement of our platform (i.e., reaching 25 m) with the other MC platforms because they are not comparable due to the difference in the nature of the signal processing used. We can however discuss what we believe allowed us to achieve a communication of 25 m with our platform. The communication over long distances in microfluidic tubing can bring two challenges: (i) high back-pressure inside this channel and (ii) high noise and interference. To address these challenges and achieve long-distance communications, we have carefully selected and programmed our microfluidic pumps and designed chemical reactions with signal processing capabilities, respectively.

To address the first challenge, we need to make sure the platform hardware can work

well at high back-pressure. Many pump types, such as membrane pumps, cannot work at very high pressure, as high pressure could cause liquids to flow backward into the pumps. In addition, the performance in keeping a stable flow rate could also be reduced at high pressure. In the design of our platform, we have selected syringe pumps to inject the chemicals. With their closed reservoir, these pumps can perform extremely well at high pressure. Moreover, we also introduced and fine-tuned a PID controller for the selected syringe pumps to stabilise the flow rate. Thus, the combination of the selection of syringe pumps and the PID control allowed our platform to maintain stable flow rates at very long distances.

The high noise and interference challenge was addressed by the design of our chemical processing functions, which is the main argument and objective of our work. The thresholding and amplification reactions implemented in the design of our platform were specifically designed to prevent signal loss due to noise and interference, allowing us to increase the communication distance without a significant impact on communication reliability.

All these elements combined together allowed us to successfully test tubing lengths up to 25 m. We can also note that this is probably not the longest distance our platform can achieve: as explained in the manuscript, we selected 25 m for the longest propagation channel to test only because this was the longest tubing length available with our supplier. While a custom-made tubing of greater length could have potentially been ordered, we believe that the scope of our paper, which was to present an experimental platform with signal processing achieved using chemical reactions, would not justify testing lengths beyond 25 m. We can also note that prior work on MC systems did not discuss their limitations on the communication distance, so we can only discuss what could have limited that distance to 4 m, and how we achieved communication over 25 m. To clarify our statement, we removed reference to the previously longest distance reported of 4 m, and to discuss which elements of our design have contributed to reaching such a long length, we have revised the manuscript as follows:

The second paragraph of Section 2.6 (page 13):

we explored the limits of our MIMIC platform by first replacing the propagation channel with the longest channel available in our lab, i.e., $L_{Ch} = 25$ m. **This tubing length conveniently corresponds to more than 10 times the average longest straight distance in the human body (head to toe).**

The fourth paragraph of the discussion (page 19):

... **In such conditions, our MIMIC platform was able to maintain a satisfying reliability, even with a distance as long as 25 m. We have also demonstrated that the use of our proposed chemical-reaction-based signal processing blocks allowed us to control the communication efficiency and the bit error rate. How-**

ever, while our platform successfully transmitted information at the highest transmission speed achievable by our platform with chemical signal processing capability, this speed is lower than the MC platform without chemical signal processing capability [31, 32]. This is because the highest transmission speed of our platform is limited by the reaction times of all the chemical signal processing functions, as well as the maximum injection speed of our 3D-printed pumps. It is important to note that higher data rates can be achieved by adapting our MIMIC platform with more powerful pumps and other kinds of faster chemical reactions.

II.12 R2 Comment 12

Given that there are 3D-printed components for the system, I suggest that the design files get attached to manuscript files if someone wants to replicate them.

Response:

We thank the reviewer for this helpful suggestion. We have added all the relevant files to 3D-print components designed for this work in the GitHub repo, which will be released upon the acceptance of this paper.

II.13 R2 Comment 13

If a pH meter was used in preparing the chemical solutions, I suggest listing it.

Response:

We thank the reviewer for this suggestion. A pH-meter was indeed used to prepare the solutions in this work. We have added the make and model of the pH meter (phenomenal PH 1100 L, VWR, USA) in the Methods section of the revised manuscript as follows:

The second paragraph of Section 4.3 (page 21):

The solutions Y, P, ThL, and Sol were all prepared from fresh PBS solution, and their pH values were adjusted by adding a small volume of HCl or NaOH (< 1 % of the total volume of PBS) until the desired pH values were reached. The pH of the solution was measured and adjusted using a phenomenal PH 1100 L pH-meter (VWR, USA).

II.14 R2 Comment 14

I would prefer to see more numbers/graphs/tables for “4.7 Communication Reliability,” giving more details about the error rate; even the Supplementary Information only discusses the measurement method, not the results.

Response:

We thank the reviewer for this observation and this suggestion. The reason why the Section “4.7 Communication Reliability” only discusses how the bit error rate (BER) was measured experimentally is that this subsection is part of the Methods section (Section 4), which is intended to present the experimental setup, methodology, chemicals and equations used to achieve the work we present in our manuscript. The results on communication reliability and error rate corresponding to the method described in Section 4.7 can be mainly found in the Results section (Section 2) of the manuscript, specifically in the Section “2.8 Communication Performance Optimization”.

In Section 2.8, we focused the discussion on how the duty cycle α could impact the BER of our platform (Fig. 6A), with the complete bit sequence used to measure the BER provided in the Supplementary Information. Indeed, we believed that the scope of this section was to demonstrate that it was possible to control and optimise the communication performance of the platform using our proposed chemical reactions. We did discuss further the cause of the errors for one specific case, $\alpha = 0.92$ through Fig. 6B to D, as we believed it is important to illustrate how these same chemical reactions could also generate errors.

As suggested by the reviewer, to discuss the communication reliability performance further, we have updated the Supplementary Information of our manuscript to add two new figures, Figs. S11 and S12, which present the complete decoded bit sequences for the measurement of the BER at different duty cycles and three corresponding types of errors, respectively. We did not include them in Sections 2.8 and 4.7 of the main text, as we do not think these results contribute to the point(s) being discussed in these sections. The text of the manuscript was however edited to add a reference to these figures in the Supplementary Information.

The first paragraph of Section 2.8 (page 16):

The complete decoded bit sequences are shown in Supplementary Fig. S11, and the typical bit errors observed during the measurement are presented in Supplementary Fig. S12.

Section 6.8 in the Supplementary Information:

The decoded bit sequences for different α are illustrated in Fig. S11, and the corresponding typical bit errors are shown in Fig. S12.

Figure S11: **Decoded bit sequences.** Illustration of the complete input and decoded bit sequences to measure the BER (Fig. 6A), with the three subsequences illustrated one after the other. Each individual bit is represented by a rectangle, and the value of each bit is illustrated by the color of the band on the graph, with light and dark grey indicating successful transmission of bit-0 and bit-1, respectively. Orange indicates the detection of a bit-1 as a bit-0, and pink indicates the detection of a bit-0 as a bit-1.

Figure S12: **Typical bit errors.** Illustration of the typical bit errors observed during the BER measurements (Figs. 6A and S11) for various duty cycles α . For each setting, a 6-bits sequence is presented with the bit error at the center of the sequence, with (A) $\alpha = 1.00$ starting at bit #54, (B) $\alpha = 0.92$ starting at bit #40, and (C) $\alpha = 0.60$ starting at bit #55. The value of each bit is illustrated by the color of the band on the graph, with light and dark grey indicating successful transmission of bit-0 and bit-1, respectively. Orange indicates the detection of a bit-1 as a bit-0, and pink indicates the detection of a bit-0 as a bit-1.

III Response to Reviewer #3

General comment

This paper describes the design and construction of a microfluidic molecular communications platform for transmitting digital signals over distances using varying concentration of sodium hydroxide. Signal shaping, thresholding, amplification, and detection by optimal (UV-Vis) spectrometer are described. Communication distance of 25 meters is achieved and 100 bits were reliably transmitted. The possible applications of molecular communication are particularly exciting in biological applications: “where the transmitter could efficiently communicate with cells and their membranes (e.g., drug delivery, DNA sensing) and the receiver could decode and read chemical signals emitted by an organism (e.g., cell culture monitoring, detection of illness-specific biomarkers).”

Given the particularly exciting applications of molecular communications in the biological context, I had a hard time connecting the system demonstrated to this context. In particular, manipulating pH/sodium hydroxide is not something that can be reasonably be done in the body or in a biologically compatible way. In this regard, communicating with nucleic acids or bio-compatible small molecules seems more aligned with the intended goal. If the intent was more on demonstrating certain fundamental principles, I wasn't sure what general ideas could be more generally extracted from the demonstrated scheme.

We would like to thank the reviewer for summarizing our paper. We have revised the paper in line with the reviewers' comments, thereby improving the contributions and the clarity of the paper accordingly. We also address here the comment on the choice of pH-based chemicals and reactions to demonstrate the proof-of-concept of our MIMIC platform. The objective of this paper is to demonstrate the principle of designing chemical reactions to process concentration signals, which has the capability to replace external electronic devices and bring bio-compatibility and non-invasivity to an MC system. Such experimental demonstration required us to choose a set of chemicals and chemical reactions that would perfectly illustrate the efficiency of our signal processing functions as well as their versatility. In regards to the reviewer's comment, this reduced our selection of chemical reactions to two categories: either reactions designed for a specific practical application with an impact in its own domain (e.g., medicine, environment science), or general reactions more designed to present our concept in a pedagogical way but with perhaps very restricted practical applications. While the former choice would indeed provide a contribution to exciting applications of MC in a domain with surely a critical interest, we believed that this choice would also restrict our audience and could even hinder the diffusion of our novelty. The main interest of pH-based reactions is not their practical applications, which as rightfully noted by the reviewer are extremely restricted, but the fact that they are universally known across many scientific domains, including chemistry,

physics, biology, medicine, and even environment science. By using chemical reactions that a large audience was familiar with, our objective is to encourage and motivate the readers to apply our overall design to their own applications. Of course, this would require then replacing the chemicals presented in this work with chemicals specific to the applications sought by readers. To clarify this decision and also to provide some examples to help readers transpose our novel design to their own applications, the manuscript has been revised as follows:

The fifth paragraph of the introduction (page 3):

To ensure that the novelties of our platform can be easily understood by different disciplines, we choose a selection of universally known pH-based reactions, based on sodium hydroxide (NaOH) and hydrochloric acid (HCl), to demonstrate and quantify the feasibility of our general chemical reaction design for electronic-free chemical signal processing functions. While these reactions might not lead to a specific application in medicine or industry, their universality can perfectly illustrate that the design principles behind our platform can be applied to any kind of reaction that would be relevant to a specific application. . . . Benefiting from the modularity of our platform, our MC transmitter and receiver are expected to operate individually for microfluidic-based applications by replacing the acid and base with application-specific chemicals and designing the corresponding chemical-reaction-based signal processing functions. . . .

The fifth paragraph of The discussion (page 16):

It is important to highlight the fact that all the experiments and equipment presented in this work are used as a strict proof-of-concept platform to validate the feasibility of chemical signal processing using the universally known acid and base-based chemical reactions as an example. However, the novelty of our MIMIC platform is not limited to these reactions and hardware, and practical applications of our MIMIC platform and its processing functions, e.g., in biology, medicine, or environmental sciences, can be achieved by adapting the chemical reactions and the platform accordingly. The versatility of our design makes it easy to replace the chemicals and the reactions used in this work with corresponding molecules specifically dedicated to the sought application. For instance, with a fluorescent dye as the signal carrier, signal suppression (i.e., used for signal shaping and thresholding) can be achieved with a black hole quencher, while signal amplification and detection can be achieved using another fluorescent probe by Förster resonance energy transfer (FRET) [51, 52]. Moreover, in order to validate our design and test its performance, several electronic devices are still needed to inject the solutions (i.e., syringe pumps) and to convert the chemical signal back into a measurable and quantifiable elec-

trical signal (i.e., the UV-Vis spectrometer). Of course, an actual application of the design of our platform would require the setup to be adapted at either the transmitter and/or receiver specific to that application. For example, to read the chemical signals strictly encoded in the acidity of a water solution extracted from the environment, the UV-Vis spectrometer could be replaced with a pH-meter. Meanwhile, to communicate with a receptor cell, one can directly remove any form of electronic sensor.

III.1 R3 Comment 1

The advancement of the state of the art in terms of quantitative measures is limited because of tradeoffs with the transmission bit rate. The authors point out the much longer communication channel and an optical readout method achieved. However, prior work using pH changes has achieved significantly larger transmission bit rate: eg ref [17] achieves 4 bits per second compared with 0.16 bits per minute in figure 4. The authors should clarify what characteristic of their system are incompatible with the faster transmission rate. The authors may also want to clarify what limited the communication distance.

Response:

We thank the reviewer for this insightful comment. We answered the different points raised by the reviewer separately in the following.

Comparison of the data rate between our platform and existing platforms

The direct comparison of the communication performance between prior work and our own platform is quite challenging. To address this comparison, it is first critical to understand the major differences between the MC systems in the prior work using pH changes and our own system described in this work our MIMIC system. While all work shared the same information carrier, the MC systems in the prior work were strictly limited to investigating how chemicals can be used for information exchange. All their signal processing functions (e.g., amplification, detection) were performed only using external electronic design, e.g., a pH-meter and an Arduino, which allowed them to increase the transmission rate as high as they wanted to. The novel design of our platform introduces here a critical change: all the signal processing is performed in the molecular domain through chemical reactions. Because of the presence of the chemical reactions used for signal processing, our platform needed to be specifically designed to take into account the reaction time of these different chemical reactions. This major difference in design results in an important difference in constraints between the platforms developed in prior work and our own MIMIC platform. Since the other platforms reported in the literature mainly focused on validating the feasibility of exchanging information via different particles, their transmission rate is not limited by chemical reactions. As a consequence, they can reach transmission rates way higher than the one we reported in this work. But because the nature of their platforms

is extremely different from the novelty of our MIMIC platform, the conditions are subsequently so different that a direct comparison on the transmission rates performances of the different platforms is complex.

Why the system is incompatible with faster transmission rates

With concentration signals processed by chemical reactions in our MIMIC platform, the communication performance is limited by two factors: (i) the nature of the hardware used to inject the chemicals, and (ii) the reaction time of the different chemical processing functions.

Consider here an MC platform working at a given flow rate. Increasing the transmission rate at this flow rate would require us to reduce the bit duration and thus the duty cycle. However, the value of the duty cycle has a lower limit, as demonstrated in Fig. 6C, below which the intensity of a signal decreases due to the existence of a transition time, i.e., the time it takes for a signal to switch between the low and high pH values. This important transition time is not only impacted by the hydrodynamics of the solution propagating in the microfluidic channel but is also impacted by the presence of the chemical reactions used to process the signal. This second point is our major difference with the other work on MC system. We have demonstrated in Supplementary Fig. S10B that this transition time is a function of the flow rate of the system. In this work, the maximum transmission rates presented were measured at the highest flow rates our pumps could achieve. So in order to reach higher transition rates, our system would first require a different set of pumps to reach higher flow rates, and further reduce the transition time that is limiting the transmission rate.

Beyond that point, it will also be essential to ensure that all chemical reactions had enough time to produce a sufficient concentration of product. At a very high flow rate, one could imagine that the solutions could leave the tubing before the chemical reaction even starts. This is another main limitation of our transmission rate, and we have investigated and briefly discussed the kinetics of the reactions in the manuscript to support this statement (Supplementary Information 3).

What limited the communication distance

We discuss here what could have limited the communication distance in our own platform. The communication over long distances in microfluidic tubing can be limited by two factors: (i) high back-pressure inside this channel and (ii) high noise and interference. To address these challenges and achieve long-distance communications, we have carefully selected and programmed our microfluidic pumps and designed chemical reactions with signal processing capabilities, respectively.

We can also note that this is probably not the longest distance our platform can achieve: as explained in the manuscript, we selected 25 m for the longest propagation channel to

test only because this was the longest tubing length available with our supplier. While a custom-made tubing of greater length could have potentially been ordered, we believe that the scope of our paper, which was to present an experimental platform with signal processing achieved using chemical reactions, would not justify testing lengths beyond 25 m. We can also note that prior work on MC systems did not discuss their limitations on communication distance, so we can only discuss what could have limited that distance to 4 m. We have revised the manuscript as follows:

The fourth paragraph of the discussion (page 19):

We evaluated the communication performance, and we explored the limits of our platform in communication, specifically to transmit and recover a text message over a long distance and at the highest speeds achievable by our hardware, which imposes higher noise and interference levels. In such conditions, our MIMIC platform was able to maintain a satisfying reliability, even with a distance as long as 25 m. We have also demonstrated that the use of our proposed chemical-reaction-based signal processing blocks allowed us to control the communication efficiency and the bit error rate. However, while our platform successfully transmitted information at the highest transmission speed achievable by our platform with chemical signal processing capability, this speed is lower than the MC platform without chemical signal processing capability [31, 32]. This is because the highest transmission speed of our platform is limited by the reaction times of all the chemical signal processing functions, as well as the maximum injection speed of our 3D-printed pumps. It is important to note that higher data rates can be achieved by adapting our MIMIC platform with more powerful pumps and other kinds of faster chemical reactions.

III.2 R3 Comment 2

If I am understanding correctly, the amplification was performed at the point of conversion from pH to Absorbance (Fig 2a), which is non-linear and exhibits a response satisfying condition (2) of the signal amplification desiderata (“in the presence of Y, the concentration value of O should not be influenced by the concentration of input Y”). Is this consistent with equation (1) which shows that Y acts catalytically to produce O?

Response:

We thank the reviewer for this comment, highlighting an issue in the clarity of our text. Due to the unclarity of our initial manuscript, it seems that the reviewer misunderstood that the non-linear amplification comes from the conversion from the concentration of O to absorbance, which they described as the “point of conversion from pH to absorbance”. However, the point of conversion from pH to absorbance involves two transformations:

conversion of the pH to a concentration of O, and the conversion of the concentration of O to the absorbance. In fact, the conversion of the pH to the concentration of O is non-linear and exhibits a response satisfying the second condition of the signal amplification desiderata, while the conversion of the concentration of O to absorbance is linear. This is therefore consistent with equation (1) showing that Y acts catalytically to produce O. The two-step conversion of pH to absorbance is illustrated in Fig. 2 below.

Figure 2: **pH to Absorbance conversion.** Illustration of the two steps used in this work to convert the pH of the solution to an absorbance value as presented in Fig. 2A.

Non-linearity in the conversion from pH to the concentration of O

The non-linearity is coming from the evolution of the concentration of O as a function of the concentration of Y during equation (1) rather than from the conversion from pH to absorbance described in Fig. 2A. The non-linearity is generated through equation (1), which satisfies the condition of the signal amplification desiderata. The amplification property of this reaction can be directly seen in the evolution of the concentration of O as a function of the concentration of Y, described by Supplementary equations (S2) and (S3) in the Supplementary Information. These two equations can be rewritten as equation (R1), which expresses the concentration of O as

$$[\text{O}] = [\text{BTB}^-] = [\text{BTB}]_{\text{Tot}} \frac{K_C 10^{\text{pH}}}{1 + K_C 10^{\text{pH}}}, \quad (\text{R1})$$

where $[\text{BTB}]_{\text{Tot}}$ is the total concentration of BTB used to prepare Amp and K_C is the equilibrium constant of the reaction described in equation (1). Because the pH value is directly related to the concentration of Y by the equation $[\text{Y}] = 10^{\text{pH}-14}$, these expressions describe the evolution of the concentration of O as a function of the concentration of Y with a sigmoidal evolution, which is similar to the sigmoid evolution shown in Fig. 2A but for the conversion from pH to absorbance. Since it is described by a sigmoid evolution, any change in the concentration of Y outside the central narrow transition range will not have

a significant impact on the evolution of the concentration O. For instance, an increase in pH from 9 to 10, corresponding to an increase in Y concentration from 10^{-5} to 10^{-4} mol/L (i.e., 10 times more Y in the solution, so a 1000% increase), will only result in an increase in O concentration by 1%.

Linearity in the conversion of the concentration O to Absorbance

We would like to clarify that the conversion from O to absorbance is linear and serves to decode the pH of transmitted signals from absorbance values as the absorbance values are the direct values obtained from our UV-Visible spectrometer. The point of conversion from O to Absorbance is itself described by the Beer-Lambert law, presented in equation (S1). The Beer-Lambert law describes the evolution of the absorbance as directly proportional to the concentration of the chemicals, here O and Amp. This means that if the conversion of the concentration of Y to the concentration of O described in equation (1) was linear, then the whole conversion from pH to absorbance would also be linear. In this work, e.g., in Fig. 2A, we only report the absorbance and later the expected pH of the solution (pH_r) because they are the values we directly have access to with our UV-Visible spectrometer.

As a conclusion, our results and observations are indeed consistent with equation (1) showing that Y acts catalytically to produce O, with the point of conversion from pH to absorbance just converting the evolution of this reaction into a value that can be measured directly by our system. To highlight the key role of the reaction between Y and Amp described by equation (1) and avoid future confusion about where the amplification takes place, we have modified the text of our revised manuscript as follows:

The third paragraph of Section 2.2 (page 7):

Through the catalytic reaction of HBTB with NaOH, the evolution of the concentration of output BTB^- with the input concentration of NaOH is non-linear and can be mathematically described by

$$[\text{BTB}^-] = [\text{BTB}]_{\text{Tot}} \frac{10^{14} K_C [\text{NaOH}]}{1 + 10^{14} K_C [\text{NaOH}]}, \quad (1)$$

with $[\text{BTB}]_{\text{Tot}}$ as the total concentration of BTB, in both acid and base forms, and K_C as the equilibrium constant of reaction (1) (see Supplementary Information 2.1). This sigmoid evolution implies that outside a narrow transition range, a large variation in $[\text{NaOH}]$ will only result in a small variation of $[\text{BTB}^-]$, resulting in the concentration evolution satisfying the second design principle of the amplification reaction. The conversion of the concentration of BTB^- into the absorbance of the BTB solution is then described by the Beer-Lambert law (Supplementary equation S1). By combining the non-linear conversion of the concentration of NaOH into the concentration of BTB^- with the linear conversion of the concentration of BTB^- into the absorbance of

the BTB solution, we obtain a non-linear conversion of NaOH into the BTB absorbance. Through our proposed mathematical transformation of the BTB absorbance into a revised pH value (pH_r), the sigmoid evolution of $[\text{BTB}^-]$ results in a pH_r evolution that spans between a low and a high pH limits. In this way, the actual pH values (pH_a), proportional to $[\text{NaOH}]$, measured outside these limits are converted to either the low or the high revised pH limits (Fig. 2A).

III.3 R3 Comment 3

Other than the above major issues, the paper is generally well-written. The figures are very helpful and nice. I found minor grammar/clarification issues:

Response:

We thank the reviewer for pointing out the grammar/clarification issues. We have addressed them accordingly.

- *line 073: awkward phrasing: “The likeliness of chemical reactions over air can be constrained drastically...”*

Response: We thank the reviewer for this comment. In the revised manuscript, we rephrased this sentence to “Considering that chemical reactions are more likely to occur in the liquid phase” as a motivation for why we developed a liquid-based MC platform rather than an air-based MC platform.

- *line 128: challenge (i) is very long and hard to understand.*

Response: We thank the reviewer for this comment. To clarify the first challenge, we rephrased this challenge in the revised manuscript as:

The first paragraph of Section 2.1 (page 4):

In this study, two critical challenges need to be addressed for the development of a liquid-based MC platform that relies on chemical reactions for signal processing: (i) a flow chemistry challenge, **jointly designing the microfluidic geometry and chemical reactions to regulate the occurrence of reactions, which enables the process of time-varying concentration signals in the molecular domain and timely communication**

- *line 190: I didn’t understand the point about compensating the variation of other pumps.*

Response: We thank the reviewer for this comment on the clarity of our text. We simply wanted to explain that in our platform the output flow rate is constant over time to avoid distortion in the signal. Thus, if Q_{solution} the flow rate of an inlet

solution pump (i.e., Y and/or P) is modified by a value ΔQ in the software, then Q_{solvent} the flow rate of the corresponding solvent pump(s) is automatically modified by a value $-\Delta Q$ so $Q_{\text{solution}} + Q_{\text{solvent}}$ is a constant over time. We do realise that our initial phrasing was making this point more complicated than it should be, and we have edited the manuscript as follows to simplify it:

The last paragraph of Section 2.1 (page 5):

To provide a baseline and to perform dilutions, a solvent Sol, made of phosphate-buffered saline (PBS) solution, **is injected into the platform.**
The Sol pump can also be used to keep the total output flow rate of the platform constant.

- *figure 1 caption: “provided for each equipment” \Rightarrow “provided for each piece of equipment”*

Response: We thank the reviewer for this suggestion. The text has been edited as requested.

- *line 322: Fig. 2c?*

Response: We thank the reviewer for pointing out this mistake. The reviewer is correct that it should be Fig. 2c, and we have corrected this in the revised manuscript.

- *line 628: There is no Fig. 5e*

Response: We thank the reviewer for pointing out this mistake. Fig. 5e should be Fig. 6a, and we have corrected this in the revised manuscript.

- *line 635: “will be hard to be decoded” \Rightarrow “will be hard to decode”*

Response: We thank the reviewer for this suggestion. The text has been edited as requested.

- *line 637: rephrase: “signal distortion induced by a twice consumption of Y”*

Response: We thank the reviewer for this suggestion. We have rephrased the sentence as follows:

The third paragraph of Section 2.8 (page 16):

To avoid this signal distortion **caused by the signal Y reacting two times with the signal suppressor P** over consecutive bit-1 signals

- *Equation (4): more explanation justifying this equation would be helpful*

Response: We thank the reviewer for this suggestion. The paragraph discussing this equation has been expanded to provide a better justification as follows:

The third paragraph of Section 2.8 (page 16):

... the bit interval T_b should be larger or equal to the sum of the width of the transmitted pulse T_e (equivalent to T_w in our model) and the emission time αT_b , hence

$$T_b \geq T_e + \alpha T_b. \quad (5)$$

Otherwise, the next signal Y will be emitted while P is still being injected. Equation (5) can be rewritten as a condition on T_b , such as

$$T_b \geq \frac{T_e}{1 - \alpha}, \quad (6)$$

where T_e can be set either directly by the injection time difference T_d in the control software or induced by the platform geometry and the selection of a path difference L_d between the inlets of Y and P greater than 0.

IV Response to Reviewer #4

General comment

The objective and key findings of the study was to demonstrate liquid based digital communication using ASCII characters, over a microfluidic channel for a distance greater than reported hitherto in the literature. The headline capability is reliable transmission of 100 bits over 25m. The digital information is encoded using chemical concentration levels and the information decoded using UV-VIS spectrometry. The system is optimized to maximise the data rate and minimise the bit error rate. The system is therefore novel, b and represents a first for this method of communication. It is however very sophisticated (complicated) in achieving a modest communication goal. The paper has 6 figures which are well constructed. There are 30 references which is less than I would expect given the number of papers published in this area.

The study is well designed, the analysis appears correct and sufficient data are presented to enable conclusions to be drawn to validate the headline claims. A limitation of the paper is a discussion on how such a system would be applied in practice and whether there are any advantages over other molecular communication (MC) systems.

*The research question is not well articulated. The paper describes **what** was done (described in both the Abstract and main text) but not **why** it was done and what is the **wider significance** (relevance) of the work itself to other workers and of the approach itself. From my perspective this is a major limitation of the paper. There is also some confusion as to the final goal: if it is to demonstrate MC at small dimensions (e.g. MC using a microfluidic platform) then what is the purpose of the validation of the technique over 25m?*

We would like to thank the reviewer for summarizing the contributions of our manuscript and their valuable suggestions in improving the quality of the manuscript. We have revised the paper in line with the reviewer's comments.

The number of the references

We thank the reviewer for pointing out that the number of references is relatively low. In the revised manuscript, we have added other 26 references, and there are now 56 references in total, including 20 on experimental MC platforms.

The objective of the paper

The objective of this paper is to design *pure concentration-based chemical* signal processing blocks through chemical reactions and develop a liquid-based MC platform to investigate their impact on communication performance. To achieve pure concentration-based chemical signal processing, we processed chemical concentration signals directly in the molecular domain through designing and implementing chemical reactions, which is

inspired by cells that rely on a complex molecular network to process biochemical signals via reacting molecules [38, 39]. In particular, we designed three chemical reactions to achieve the signal shaping, signal thresholding, and signal amplification and detection functions. We applied these three reactions in different regions of a microfluidic system and finely tuned the microfluidic geometry to ensure that the designed signal processing functions are performed appropriately. To make our MIMIC platform easily reachable by the entire MC community as well as other communities such as chemistry and biology, we used the universally known pH-based reactions and encoded information into the pH of a solution. In addition, we evaluated the communication performance of our MIMIC platform and studied how the proposed reactions improved communication quality. Although the pH-based reactions used in this work are just a proof-of-concept of our methodology, the principles behind the methodology and platform can lead to various applications.

The motivation of the paper

The motivation for performing signal processing functions over chemical signals is to develop a novel method for chemical signal processing that can meet the biocompatible, non-invasive, and size-miniaturized requirements of applications in various domains, e.g., in medicine, biology, and environment science. This motivation is further supported by the fact that the signal processing functions of existing MC platforms are performed before molecule release or after molecule reception using external electronic devices. As these existing MC platforms focused on validating the feasibility of exchanging information via various molecule types, it is acceptable for them to use electronic devices to validate this feasibility. However, this utilization of electronic devices for electrical signal processing functions will hinder the unleashing of the potential of MC for interdisciplinary applications, a major problem that our chemical-reaction-based signal processing can solve. In the revised manuscript, we clarified our objective and motivation further as below.

The third paragraph of the introduction (page 2):

... It is important to note that the electrical devices here are integrated with these macroscale instruments to perform signal processing functions (e.g., encoding-decoding [22] and detection algorithms [18, 22, 23, 25, 27]) based solely on electrical bit signals, with the aim of ensuring successful information transmission after propagation. Although it is acceptable to use electronic devices for signal processing when focusing on the information exchange based on chemical molecules, the utilization of electronic devices for signal processing functions can hardly meet the biocompatible, non-invasive, and size-miniaturized requirements of most microscale/nanoscale biochemical applications, e.g., tissue engineering, targeted drug delivery, and immune system enhancement [37]. Thus, it is important to shift from electrical signal processing to chemical signal processing and design a family of pure chemical

concentration signal processing (e.g., pulse shaping and thresholding) blocks that operate in real-time for signal generation, detection, amplification, etc. Nevertheless, the experimental implementation of these basic signal processing functions over chemical signals, especially at the microscale/nanoscale, has been ignored so far.

The applications, advantages, and wider significance of our platform

We thank the reviewer for this comment highlighting the lack of clarity in the presentation of the applications sought by our platform, its advantages over existing MC systems, and the wider significance of our platform. We will address these different points here to demonstrate the pertinence of our work.

- **Applications:** The unique design of our platform, where chemical signals are processed via a series of chemical reactions instead of electronic devices, considerably extends the potential applications of our platform as compared to the other MC platforms previously reported in the literature. Going beyond modest information transmission, the ability to process chemical signals without the need of electronic devices allows the chemical signal processing blocks designed in our MC platform to be used in biocompatible and non-invasive applications (e.g., tissue engineering, targeted drug delivery, cancer detection and treatment, immune system enhancement), a range of applications that the more simple and straightforward designs of others platforms cannot achieve. The modularity offered by our design also opens up the path to applications for the individual signal processing blocks presented in our work, since each of these blocks works independently of each other. As demonstrated by our results, the signal amplification and detection reaction can be utilised to enhance the sensitivity of detectors/sensors and even to allow the detection of an invisible chemical signal.
- **Advantages:** Compared to other MC platforms, our platform presents multiple unique advantages that would facilitate its application and adoption in different societies (e.g., MC, chemistry, medicine). As explained before, the biocompatibility and non-invasiveness provided by our unique design is by itself a key advantage of our MC platform as compared to other platforms which offer an alternative to electronic signal processing. But this is not the only advantage, and we can also mention the modularity of our platform, where chemical signal processing blocks can be added, removed, moved, or modified, as well as its plug-and-play design. By using simple microfluidic tubing and connectors, the entire platform can be totally re-designed and re-purposed in a short time to fit any experiment or application. Moreover, by providing all the files to 3d print our pumps as well as the code of our control software along with our manuscript, our platform is characterised by a low cost but

reliable design that can be reproduced by anyone, with notably a price per pump of about 75 euros while the price of commercial pumps is often way above 1000 euros.

- **Wider significance:** Apart from the different applications mentioned in the first paragraph, the method presented in this paper, i.e., processing chemical signals through designing chemical reactions, can help MC researchers implement their proposed signal processing functions (e.g., various detection algorithms) and produce new and more complex signal processing blocks to generate user-defined time-varying chemical signals. These include but are not limited to the oscillator, memory, feedback control, and other filtering blocks, the successful implementation of these signal processing blocks in a biologically compatible way can further facilitate MC towards practical applications. For example, the oscillator block can be used to mimic cell fluctuating growth conditions for microbial growth studies [R4], while the filtering block (e.g., a low/band/high-pass filter) can regulate the drug release to targeted sites in drug delivery applications only when a biomarker is periodically detected.

[R4] Frank, P. *et al.* Autonomous Integrated Microfluidic Circuits for Chip-Level Flow Control Utilizing Chemofluidic Transistors. *Adv. Funct. Mater.* **27**, 1700430 (2017).

To better clarify the applications, advantages, and wider significance of our platform, we have edited the manuscript as follows:

The fifth paragraph of the introduction (page 3):

Benefiting from the modularity of our platform, our MC transmitter and receiver are expected to operate individually for microfluidic-based applications by replacing the acid and base with application-specific chemicals and designing the corresponding chemical-reaction-based signal processing functions.

The discussion (page 16):

... the chemical dimension of these platforms is strictly restricted to the use of molecules as a signal carrier, and all the necessary signal processing functions of these platforms are entirely performed in external electronic devices. This lack of chemical concentration signal processing would hinder the evolution of these platforms to practical applications. Motivated by the previous theoretical MC works that revealed the chemical signal processing capabilities of chemical reactions [40-44], we report in this paper the first experimental implementation of a MIMIC platform that achieves real-time signal shaping, thresholding, amplification, and detection functions for chemical concentration signals in the molecular domain.

A key aspect of performing signal processing in the molecular domain is the versatility of these functions, which theoretically can be applied to any molecular signal carrier and to any form of encoding electrical bit signals to chemical

signals. ... Correspondingly, processing chemical concentration signals will therefore either modify the physico-chemical properties of a molecule or modify the composition of a solution. To present the proof-of-concept of our design and illustrate clearly the versatility of our design, the experiments presented in this work for our MIMIC platform rely on an extrinsic encoding of the information into the concentration of a base solution, NaOH. Thus, all chemical reactions presented in this work process NaOH concentration signals based either on modifying the composition of a NaOH solution (i.e., acid-base reactions for signal shaping and thresholding) or on the change of the physico-chemical properties of a pH indicator by the presence or absence of NaOH (i.e., signal amplification and detection). Although other MC studies already investigated encoding signals in acid and base solutions [22, 23], our MIMIC platform is the first reported platform that uses reactions to process signals in real-time, whereas other platforms left all the processing tasks to computers and micro-controllers.

The advancement and versatility of our platform are also reflected in the selection of the hardware. The previously published studies on acid and base encoding in MC all used a pH-meter to extract information from concentration variations, while our MIMIC platform used a UV-Visible spectrometer. This choice was made to better illustrate the versatility of our design: while a pH-meter is a convenient solution to quantify the properties of acid-base solutions, it cannot be easily applied to other chemical reactions due to its specificity. By contrast, while a UV-Visible spectrometer is not capable of directly detecting changes in acidity, we have successfully demonstrated how chemical reactions can convert an invisible chemical signal into a visible chemical signal. The capability of our chemical-reaction-based signal processing functions to change the nature of the signal carrier brings a versatility that allows us to adapt the signal to the detector type at the receiver, and combined with the natural versatility of the UV-Vis spectrometer proves the versatility of our MIMIC platform.

We evaluated the communication performance, and we explored the limits of our platform in communication, specifically to transmit and recover a text message over a long distance and at the highest speeds achievable by our hardware, which imposes higher noise and interference levels. In such conditions, our MIMIC platform was able to maintain a satisfying reliability, even with a distance as long as 25 m. We have also demonstrated that the use of our proposed chemical-reaction-based signal processing blocks allowed us to control the communication efficiency and the bit error rate. However, while our platform successfully transmitted information at the highest transmission

speed achievable by our platform with chemical signal processing capability, this speed is lower than the MC platform without chemical signal processing capability [31, 32]. This is because the highest transmission speed of our platform is limited by the reaction times of all the chemical signal processing functions, as well as the maximum injection speed of our 3D-printed pumps. It is important to note that higher data rates can be achieved by adapting our MIMIC platform with more powerful pumps and other kinds of faster chemical reactions.

It is important to highlight the fact that all the experiments and equipment presented in this work are used as a strict proof-of-concept platform to validate the feasibility of chemical signal processing using the universally known acid and base-based chemical reactions as an example. However, the novelty of our MIMIC platform is not limited to these reactions and hardware, and practical applications of our MIMIC platform and its processing functions, e.g., in biology, medicine, or environmental sciences, can be achieved by adapting the chemical reactions and the platform accordingly. The versatility of our design makes it easy to replace the chemicals and the reactions used in this work with corresponding molecules specifically dedicated to the sought application. For instance, with a fluorescent dye as the signal carrier, signal suppression (i.e., used for signal shaping and thresholding) can be achieved with a black hole quencher, while signal amplification and detection can be achieved using another fluorescent probe by Förster resonance energy transfer (FRET) [51, 52]. Moreover, in order to validate our design and test its performance, several electronic devices are still needed to inject the solutions (i.e., syringe pumps) and to convert the chemical signal back into a measurable and quantifiable electrical signal (i.e., the UV-Vis spectrometer). Of course, an actual application of the design of our platform would require the setup to be adapted at either the transmitter and/or receiver specific to that application. For example, to read the chemical signals strictly encoded in the acidity of a water solution extracted from the environment, the UV-Vis spectrometer could be replaced with a pH-meter. Meanwhile, to communicate with a receptor cell, one can directly remove any form of electronic sensor.

For all potential applications, our low-cost plug-and-play platform allows for rapid testing of iterative design, optimization, and validation of new application-related chemical reactions and microfluidic geometry. . . . In this perspective, by proposing a versatile, low-cost, and adaptable design capable of replacing all electronic devices with biocompatible and non-invasive signal processing functions in the molecular domain, our MIMIC platform is a major step toward the multidisciplinary applications promised by MC, e.g., in medicine, biology,

chemistry, and environmental sciences.

The information transmission over a 25 m channel

We thank the reviewer for this helpful comment on the clarity of our motivation. We agree with the reviewer that our MC platform is designed to demonstrate communication at the microscale, with signals processed at the molecular level and the propagation performed in the tubing of diameter 0.75 mm. But it is important to note that exploring long distances of propagation is critical, even for microscale MC testbeds such as ours. Even in nature, MC is not limited to distances of a few micrometers. For example, communication in the human body between the brain and the cells is performed typically at lengths over a meter long. It is also noted that the experimental development of MC platforms requires a prototyping phase that can hardly be performed entirely at the microscale. Indeed, while a microfluidic chip can be built to dimensions close to the ones found in a living organism, they completely lack modularity, and any tiny change in the design would always result in the necessity to build a new chip. In comparison, our microfluidic MC testbed can be entirely modified in a few seconds even to reflect major changes in design, with a plug-and-play property provided by the use of microfluidic tubings. This is why many experimental MC platforms reported in the literature are found with typical propagation lengths in the order of a few meters [17, 21, 23].

As a consequence, the development of our novel MC platform design, where all signal processing functions are entirely achieved through chemical reactions, should also demonstrate its performance over long distances. This can not only demonstrate the feasibility of communication over the longest distances found in a human body but also prove that this design can be used by anyone who wants to integrate it into their MC development platforms where the architecture already includes propagation over several meters. This motivated us to explore the limitation of our MIMIC platform, especially its communication reliability performances over long distances and at high speeds. The choice of measuring the communication performance over a distance as long as 25 m served several purposes. Not only did it conveniently corresponds to more than 10 times the longest distances one could find in the human body, from the top of the head to the toes, but it also corresponds to the longest tubing length available with our supplier. Since all sources of potential error, noise, and interference are proportional to the propagation distance, this test allowed us to conclude that our design can be safely used to build a prototyping MC testbed of any size and demonstrate to the MC society that our design can be adapted to the distances typically used in other MC platforms. It also demonstrates that our design can be miniaturized at small dimensions, since there is no impact on the system size. Here, we do not investigate further how miniaturization can be achieved, as this would be a pure microfluidic flow chemistry challenge and we believe that the scope of our paper is only to demonstrate how signals can be processed in the molecular domain, and how

it can be integrated in a proof-of-concept prototyping MC platform. To clarify how the longest distance we have tested is related to the motivation of our work, we have revised the manuscript as follows:

The fifth paragraph of the introduction (page 3):

... Using these reactions, we experimentally demonstrate that our designed MIMIC platform can reliably transmit bit signals with a low bit error rate (BER), at the highest transmission speeds achievable by our hardware and over extremely long distances, testing tubing up to 25 m long. ...

The first and second paragraphs of Section 2.6 (page 13):

To explore the limits of the communication performance of our MIMIC platform, we investigated the communication efficiency for transmissions at the highest speeds achievable by our system and over a long distance ... This tubing length conveniently corresponds to more than 10 times the average longest straight distance in the human body (head to toe). ...

The fourth paragraph of the discussion (page 19):

We evaluated the communication performance, and we explored the limits of our platform in communication, specifically to transmit and recover a text message over a long distance and at the highest speeds achievable by our hardware, which imposes higher noise and interference levels. In such conditions, our MIMIC platform was able to maintain a satisfying reliability, even with a distance as long as 25 m.

IV.1 R4 Comment 1

Title: The title is descriptive, but ‘Realizations...’ does not read well. I suggest this word is omitted altogether in future revisions.

Response:

We thank the reviewer for this suggestion. We have removed the word “Realizations” from the title of our manuscript, and the current title is “Signal Processing via Chemical Reactions for a Microfluidic Molecular Communication System”.

IV.2 R4 Comment 2

Abstract: This needs a clear statement of the research question, its significance and relevance to other MC workers and the wider community.

Response:

We thank the reviewer for this helpful comment. We responded to the different points separately in the following.

The research question

Following the previous answer to the general comment, the research question we explored in this paper is how to perform signal processing functions over the molecular domain and develop bio-compatible and non-invasive MC platforms. In existing experimental MC platforms, signal processing functions are performed via electronic devices. However, the utilization of electronic devices for signal processing can hardly meet the biocompatible, non-invasive, and size-miniaturized requirements of some microscale/nanoscale biochemical applications. To explain this, we revised the abstract in the manuscript as follows:

The abstract:

Signal processing over the molecular domain is critical for analyzing, modifying, and synthesizing chemical signals in molecular communication (MC) systems. **However, the lack of chemical signal processing blocks and the wide use of electronic devices to process electrical signals in existing MC platforms can hardly meet the biocompatible, non-invasive, and size-miniaturized requirements of applications in various domains, e.g., in medicine, biology, and environment sciences. To tackle this,** we designed and constructed a liquid-based microfluidic molecular communication (MIMIC) platform for **performing chemical concentration signal processing and** digital signal transmission over distances. . . .

The significance and relevance to MC workers and the wider community

We agree with the reviewer that the abstract needs a statement of the significance and relevance of our MIMIC platform to MC and other societies. This work demonstrates how chemical signals can be processed over the molecular domain in a biocompatible and non-invasive way, which is absent in the prior MC works due to their use of electronic devices for signal processing. The novelty presented through the design of our platform can enable the implementation of signal processing units in biological settings and then unleash the potential of MC for biocompatible applications in different disciplines, such as biology, medicine, or environmental sciences. This can be achieved by replacing the chemicals and the reactions used in this work with corresponding molecules specifically dedicated to the sought application. Moreover, the method of designing chemical reactions to process time-varying concentration signals presented in this work could help MC researchers implement their proposed signal processing functions (e.g., various detection algorithms) and produce other new and more complex signal processing blocks. The development of such applications can be achieved through the detailed methodology, equations, software, and 3D printer files provided with this work, making it easy for MC workers and the wider community to reproduce and adapt our design by simply replacing the chemicals

and reactions to the one specifically dedicated to the sought application, or by developing and implementing their own proposed signal processing functions (e.g., various detection algorithms) and produce new and more complex signal processing blocks. To clarify this, we revised the abstract as follows:

The abstract:

... This platform is further optimized to maximise the data rate while minimising the communication error. **The novel methodology for real-time chemical signal processing presented in this work can enable the implementation of signal processing units in biological settings and then unleash its potential for interdisciplinary applications.**

IV.3 R4 Comment 3

***Introduction:** A survey of the literature is given but there are some important omissions which are relevant to the experiments undertaken (e.g. McGuinness et al, ‘Experimental and Analytical Analysis of Macro-Scale Molecular Communications within Closed Boundaries’, IEEE Trans. on Molecular, Biological and Multi-scale Communications, 5, 1 (2019))*

*The paper claims (lines 94-97) that ‘To achieve the goal ... it is fundamentally important to design a family of pure chemical concentration, signal processing ... blocks that operate in real-time...’ The paper however does not say why this is **fundamentally** important.*

Response:

We thank the reviewer for bringing this reference to our attention and pointing out that we did not explain why designing a family of pure chemical concentration signal processing blocks is fundamentally important. We responded to these points separately in the following.

The new reference

In the revised manuscript, we have added this reference ([21] in the revised manuscript) to the introduction and also added other 12 papers which reported gas-based or liquid-based MC platforms. Please refer to our response to the general comment on page 40.

Why it is fundamentally important

Designing a family of pure chemical concentration signal processing blocks is fundamentally important because the electronic devices, which were utilised for signal processing in existing experimental MC platforms, are difficult to miniaturise to nano/micrometer-scale and can hardly meet the biocompatible requirement of MC-promised applications. Although it is acceptable to use electronic devices to investigate the feasibility of using chemical molecules to exchange information, the utilization of electronic devices to process digital signals and a lack of chemical concentration signal processing blocks would hinder

the unleashing of the potential of MC for interdisciplinary applications. To make this clear, we revised the manuscript as follows:

The third paragraph of the introduction (page 2):

... It is important to note that the electrical devices here are integrated with these macroscale instruments to perform signal processing functions (e.g., encoding-decoding [22] and detection algorithms [18, 22, 23, 25, 27]) based solely on electrical bit signals, with the aim of ensuring successful information transmission after propagation. Although it is acceptable to use electronic devices for signal processing when focusing on the information exchange based on chemical molecules, the utilization of electronic devices for signal processing functions can hardly meet the biocompatible, non-invasive, and size-miniaturized requirements of most microscale/nanoscale biochemical applications, e.g., tissue engineering, targeted drug delivery, and immune system enhancement [37]. Thus, it is important to shift from electrical signal processing to chemical signal processing and design a family of pure chemical concentration signal processing (e.g., pulse shaping and thresholding) blocks that operate in real-time for signal generation, detection, amplification, etc. Nevertheless, the experimental implementation of these basic signal processing functions over chemical signals, especially at the microscale/nanoscale, has been ignored so far.

IV.4 R4 Comment 4

Results and Methodology: *The figures are clear, well presented and of high quality. The methodology is described in detail and perhaps this section could be shortened. The headline claims of the study of are validated by the information presented. Proof of principle of using liquid chemical concentration and flow chemistry to transmit information is therefore demonstrated. System reproducibility by other workers is possible with the information provided. However, the system described is complex and very sophisticated for modest information transmission. Unless the authors can demonstrate some key and unique advantages using their approach over other approaches (e.g. using gas concentrations) I believe that adoption by other workers will be very limited.*

Response:

We thank the reviewer for this comment as well as his/her opinion on the quality of our work. We answered the three comments raised here separately in the following.

The methodology perhaps could be shortened

We believe that all the methodology provided is required to ensure the system's reproducibility. As positively commented by Reviewer 1 and Reviewer 4 themselves, the current

presentation of the methodology supports our results by demonstrating the careful and precise development of our design as well as the data collection, and provides all the information needed to readers in order to exactly reproduce the experiments if needed. This is especially important since we present here the proof-of-concept of an MC platform designed as a prototyping testbed that can be used by different scientific communities. Reducing the methodology part could have a negative impact on the clarity of the design, and makes it harder for the reader to reproduce.

The system described is complex and very sophisticated

It is indeed correct that our platform is more sophisticated and can even appear more complex than most of the MC platforms previously reported in the literature, but the reviewer seems to have missed the major difference in the purpose of each platform. Unlike prior work reported in the literature with a simple experimental setup, the purpose of our platform is not limited to demonstrate the feasibility of modest information transmission using chemicals as the signal carrier, but instead to demonstrate that chemical signals can be directly processed in the molecular domain through chemical reactions. Achieving this goal does require an extra level of sophistication which we are the first to report experimentally. Despite this apparent complexity, we believe we have made our work accessible and reproducible for readers, since we have presented thoroughly in our manuscript not only all the methodology and calculations that need to be used, but also the files to 3D print our open-source syringe pumps and the code of our control software. This will allow readers to reproduce our results and adapt our design to different sets of chemical reactions. In comparison to our low-cost platform, with pumps that can be produced for less than \$100 each, the much more simple MC platforms reported in prior work will not be capable of processing chemical signals in the molecular domain without a thorough revision of their designs.

Key and unique advantages

To date, all the approaches reported in the literature, including approaches where the signal is encoded into gas concentrations, are only addressing the modest and straightforward application of encoding and transmitting information based on chemical molecules. As a consequence, all these MC platforms are ignoring the challenge of processing chemical signals directly in the molecular domain, and instead heavily rely on external electronic devices to process electrical signals before encoding and decoding. However, in an application where either the transmitter or receiver is replaced by an organism (e.g., cell, organ, living being) or any non-computer-controlled transceiver, it is no longer possible to ignore the importance of processing chemical signals in the molecular domain. Unlike prior work, the purpose of our platform is not to just simply encode and decode a chemical signal but to process that signal using a set of carefully chosen chemical reactions. In the scope of this paper, we focused on experimentally demonstrating the proof-of-concept of this

approach, which is a novelty in the development of MC platforms. To better achieve this demonstration and explore the performance and limits of our design, we made the decision to take a simple yet well-known set of chemical reactions, as well as use pumps and a UV-Visible spectrometer to have full control of the injection and detection of our platform. The simple information transmission experiments we present in our work are only used to present the performance of our platform by using protocols and metrics commonly used in the MC community.

We believe that through this work we have successfully demonstrated, using our simple set of chemical reactions, that it is possible to process a chemical signal directly in the molecular domain. This demonstration will open the fields of MC researchers to design and validate other chemical signal processing blocks by adapting our MIMIC platform, and open the path to future experimental development and research where modest information communication will no longer be enough, and where a non-invasive bio-compatible MC design pioneered by our system can become a standard approach. We have revised the manuscript to clarify the objective sought by this work and further highlight the key and unique advantages of our approach over other approaches. The manuscript is revised as follows:

The discussion (page 16):

... the **chemical dimension** of these platforms is strictly restricted to the use of molecules as a signal carrier, and all the necessary signal processing functions of these platforms are entirely performed in external electronic devices. This lack of chemical concentration signal processing would hinder the evolution of these platforms to practical applications. Motivated by the previous theoretical MC works that revealed the chemical signal processing capabilities of chemical reactions [40-44], we report in this paper the first experimental implementation of a MIMIC platform that achieves real-time signal shaping, thresholding, amplification, and detection functions for chemical concentration signals **in the molecular domain**.

A key aspect of performing signal processing in the molecular domain is the versatility of these functions, which theoretically can be applied to any molecular signal carrier and to any form of encoding electrical bit signals to chemical signals. ... Correspondingly, processing chemical concentration signals will therefore either modify the physico-chemical properties of a molecule or modify the composition of a solution. To present the proof-of-concept of our design and illustrate clearly the versatility of our design, the experiments presented in this work for our MIMIC platform rely on an extrinsic encoding of the information into the concentration of a base solution, NaOH. Thus, all chemical reactions presented in this work process NaOH concentration signals based ei-

ther on modifying the composition of a NaOH solution (i.e., acid-base reactions for signal shaping and thresholding) or on the change of the physico-chemical properties of a pH indicator by the presence or absence of NaOH (i.e., signal amplification and detection). Although other MC studies already investigated encoding signals in acid and base solutions [22, 23], our MIMIC platform is the first reported platform that uses reactions to process signals in real-time, whereas other platforms left all the processing tasks to computers and micro-controllers.

We evaluated the communication performance, and we explored the limits of our platform in communication, specifically to transmit and recover a text message over a long distance and at the highest speeds achievable by our hardware, which imposes higher noise and interference levels. In such conditions, our MIMIC platform was able to maintain a satisfying reliability, even with a distance as long as 25 m. We have also demonstrated that the use of our proposed chemical-reaction-based signal processing blocks allowed us to control the communication efficiency and the bit error rate. ...

... However, the novelty of our MIMIC platform is not limited to these reactions and hardware, and practical applications of our MIMIC platform and its processing functions, e.g., in biology, medicine, or environmental sciences, can be achieved by adapting the chemical reactions and the platform accordingly. The versatility of our design makes it easy to replace the chemicals and the reactions used in this work with corresponding molecules specifically dedicated to the sought application. For instance, with a fluorescent dye as the signal carrier, signal suppression (i.e., used for signal shaping and thresholding) can be achieved with a black hole quencher, while signal amplification and detection can be achieved using another fluorescent probe by Förster resonance energy transfer (FRET) [51, 52]. ...

For all potential applications, our low-cost plug-and-play platform allows for rapid testing of iterative design, optimization, and validation of new application-related chemical reactions and microfluidic geometry. ... In this perspective, by proposing a versatile, low-cost, and adaptable design capable of replacing all electronic devices with biocompatible and non-invasive signal processing functions in the molecular domain, our MIMIC platform is a major step toward the multidisciplinary applications promised by MC, e.g., in medicine, biology, chemistry, and environmental sciences.

IV.5 R4 Comment 5

***Discussion:** The paper does not clearly explain the implications of the study for the field and its potential future applications. Discussion of the limitations of their technique in comparison with other published MC systems is lacking. The paper claims that the work is a ‘major step toward promised MC applications...’ (Line 742), but this conclusion is not supported by the paper it would be unlikely in my view, on the basis of what has been reported in the paper that MC using this approach would be adopted.*

Response:

We thank the reviewer for this helpful and constructive comment. We responded to the three points separately in the following.

The implications of the study and future applications

All the previously reported MC platforms were processing the signals either before encoding digital bits to chemical signals or after decoding chemical signals to digital bits. While this approach allows for a simplification of the platform designs, it strictly limits the applications of these designs to applications where both the transmitter and the receiver are computer-controlled or at least controlled by external electronic devices. Thus, it prevents any form of non-invasive bio-compatible applications, where either or both transceivers are replaced with organisms. The novelty brought by our MC platform and our results is that signal processing functions can be performed in the molecular domain. Our work brings to the field of MC the capability of designing experimental platforms that can be bio-compatible, thus allowing the development of solutions for bio-compatible applications (e.g., drug delivery, automated cell culture). To better clarify the implications of our work for the field and to list potential future applications, we have edited the manuscript as follows:

The first paragraph of The discussion (page 16):

... the **chemical dimension of these platforms is strictly restricted to the use of molecules as a signal carrier, and all the necessary signal processing functions of these platforms are entirely performed in external electronic devices. This lack of chemical concentration signal processing would hinder the evolution of these platforms to practical applications. Motivated by the previous theoretical MC works that revealed the chemical signal processing capabilities of chemical reactions [40-44], we report in this paper the first experimental implementation of a MIMIC platform that achieves real-time signal shaping, thresholding, amplification, and detection functions for chemical concentration signals in the molecular domain.**

The fifth paragraph of The discussion (page 16):

It is important to highlight the fact that all the experiments and equipment

presented in this work are used as a strict proof-of-concept platform to validate the feasibility of chemical signal processing using the universally known acid and base-based chemical reactions as an example. However, the novelty of our MIMIC platform is not limited to these reactions and hardware, and practical applications of our MIMIC platform and its processing functions, e.g., in biology, medicine, or environmental sciences, can be achieved by adapting the chemical reactions and the platform accordingly. The versatility of our design makes it easy to replace the chemicals and the reactions used in this work with corresponding molecules specifically dedicated to the sought application. For instance, with a fluorescent dye as the signal carrier, signal suppression (i.e., used for signal shaping and thresholding) can be achieved with a black hole quencher, while signal amplification and detection can be achieved using another fluorescent probe by Förster resonance energy transfer (FRET) [51, 52]. Moreover, in order to validate our design and test its performance, several electronic devices are still needed to inject the solutions (i.e., syringe pumps) and to convert the chemical signal back into a measurable and quantifiable electrical signal (i.e., the UV-Vis spectrometer). Of course, an actual application of the design of our platform would require the setup to be adapted at either the transmitter and/or receiver specific to that application. For example, to read the chemical signals strictly encoded in the acidity of a water solution extracted from the environment, the UV-Vis spectrometer could be replaced with a pH-meter. Meanwhile, to communicate with a receptor cell, one can directly remove any form of electronic sensor.

Discussion of the limitations of our system

In the following, we present the limitations of our platform compared to other published MC systems. As addressed in our answer to **Comment 4**, the purpose of our platform is not limited to modest information communication, but to present our novel method to process the time-varying chemical signals directly in the molecular domain using chemical reactions. As a consequence, our platform has a transmission rate limited by the reaction time of chemical reactions. Since other published platforms rely only on external electronic devices to process their signals, their transmission rates do not face the same limitation and can achieve larger rates. However, the critical change introduced by our design, i.e., performing signal processing functions in the molecular domain, opens our platform the path to non-invasive bio-compatible applications. Because of their simple design relying on external electronic devices, other published platforms are not capable of achieving the same non-invasive and bio-compatible properties and are therefore application-limited. We have edited the manuscript to add a discussion on the major difference between our MC platform and the previously reported platform, highlighting how the differences in

application introduce different specific limitations.

The fourth paragraph of the discussion (page 19):

We evaluated the communication performance, and we explored the limits of our platform in communication, specifically to transmit and recover a text message over a long distance and at the highest speeds achievable by our hardware, which imposes higher noise and interference levels. In such conditions, our MIMIC platform was able to maintain a satisfying reliability, even with a distance as long as 25 m. We have also demonstrated that the use of our proposed chemical-reaction-based signal processing blocks allowed us to control the communication efficiency and the bit error rate. However, while our platform successfully transmitted information at the highest transmission speed achievable by our platform with chemical signal processing capability, this speed is lower than the MC platform without signal processing capability [31, 32]. This is because the highest transmission speed of our platform is limited by the reaction times of all the chemical signal processing functions, as well as the maximum injection speed of our 3D-printed pumps. It is important to note that higher data rates can be achieved by adapting our MIMIC platform with more powerful pumps and other kinds of faster chemical reactions.

The adoption of our approach

We do believe that, through the design of our MC platform and our results on its communication performance, the claim made in our paper is well supported. By mimicking the molecular-based communication methods found in nature, the majority of envisioned ultimate applications for MC involve direct communication with a living organism, e.g., drug delivery, blood sugar level monitoring, automated cell culture, and cancer detection. However, the development of MC platforms to achieve these applications requires perfect compatibility between the platform and the final host [15]. Signal processing functions are essential in every communication paradigm, and so far all the previously published MC platforms are relying on external electronic devices to achieve signal processing functions. But such external devices prevent these platforms from being bio-compatible, thus preventing further development of these platforms to tackle the ultimate applications for MC. The MC design we have presented and demonstrated in this work introduces the use of processing chemical signals in the molecular domain to control the information flow. This novelty removes a major limitation in the adaptation of MC platforms to bio-compatible applications. Therefore, this makes our design a major step toward promised MC applications, as claimed in our manuscript. We have revised our text to better support that claim as follows:

The last paragraph of The discussion (page 16):

For all potential applications, our low-cost plug-and-play platform allows for

rapid testing of iterative design, optimization, and validation of new application-related chemical reactions and microfluidic geometry. . . . **In this perspective, by proposing a versatile, low-cost, and adaptable design capable of replacing all electronic devices with biocompatible and non-invasive signal processing functions in the molecular domain, our MIMIC platform is a major step toward the multidisciplinary applications promised by MC, e.g., in medicine, biology, chemistry, and environmental sciences.**

IV.6 R4 Comment 6

Minor changes: Apart from the change in title (above) I can see few minor corrections that are needed.

Response:

We thank the reviewer for bringing the existence of minor corrections to our attention. We have carefully checked the manuscript for minor revisions.

Reviewers' Comments:

Reviewer #1:

Remarks to the Author:

I can repeat from my initial review that I liked the concepts presented, which are novel and have the potential to push the research in molecular communications further. Many of the issues raised have been discussed appropriately by the authors.

No paper is perfect and every reviewer also has a subjective bias. Pushing this aside, I acknowledge the significant improvement of the manuscript and the value of the paper for the research community. I would, of course, also love to discuss further with the authors about follow-up experiments to better shape the system towards application in, e.g., the internet of bio nano things.

I suggest to accept the manuscript at this stage.

Reviewer #2:

Remarks to the Author:

I think the authors did a decent job addressing the comments; the manuscript reads much better to me right now. I only have two minor comments:

- 1) You might want to avoid using the words "novel," "new,"..etc., as recommended in the formatting instructions (which I recommend re-reading).
- 2) As stated in the manuscript, HCl & NaOH are used for their understandability while other systems can benefit from more complex reactions; Based on this statement, if feasible, it would add value to highlight any example reactions that would benefit from the system in any practical field.

Reviewer #3:

Remarks to the Author:

I believe that the authors have adequately addressed my concerns from the original review.

Reviewer #4:

Remarks to the Author:

I am impressed by the detailed and careful responses of the authors to the comments made by each of the reviewers including those made by me. The changes made by the authors provide clarity, additional information and are a genuine attempt to satisfy the reviewers and the original paper is much improved as a result.

**Response to Reviewers' Comments for
Manuscript Paper ID NCOMMS-23-12790A**

Real-time signal processing via chemical reactions for a microfluidic
molecular communication system

Addressed Comments for publication to NATURE COMMUNICATIONS by the authors

October 10, 2023

Dear Reviewers,

We would like to thank the reviewers for reviewing our paper again. We are also indebted to the reviewers for their positive feedback and minor comments. We have addressed all the comments and updated the manuscript. To enhance the legibility of this response letter, all the reviewers' comments are typeset in *italic font* and blue, and our responses are written in plain font. Rephrased sentences are typeset in red.

Yours sincerely,
The authors

I Response to Reviewer #1

I can repeat from my initial review that I liked the concepts presented, which are novel and have the potential to push the research in molecular communications further. Many of the issues raised have been discussed appropriately by the authors.

No paper is perfect and every reviewer also has a subjective bias. Pushing this aside, I acknowledge the significant improvement of the manuscript and the value of the paper for the research community. I would, of course, also love to discuss further with the authors about follow-up experiments to better shape the system towards application in, e.g., the internet of bio nano things.

I suggest to accept the manuscript at this stage.

Response:

We would like to thank the reviewer for the positive feedback and for suggesting to accept the current manuscript.

II Response to Reviewer #2

I think the authors did a decent job addressing the comments; the manuscript reads much better to me right now. I only have two minor comments.

Response:

We thank the reviewer for recognising our efforts to revise the manuscript and prepare the response letter. We respond to the two minor comments in the following.

Comment 1:

You might want to avoid using the words “novel,” “new,”..etc., as recommended in the formatting instructions (which I recommend re-reading).

Response:

We thank the reviewer for this comment. We have removed these words in the revised manuscript.

Comment 2:

As stated in the manuscript, HCl & NaOH are used for their understandability while other systems can benefit from more complex reactions; Based on this statement, if feasible, it would add value to highlight any example reactions that would benefit from the system in any practical field.

Response:

We thank the reviewer for this constructive comment. It is noted that we have highlighted an example reaction that would benefit from our MIMIC platform in the Discussion section, i.e., the Förster resonance energy transfer (FRET) based reactions, which have practical applications in fields such as medicine, biotechnology, and drug discovery. In this example, the signal carrier Y can be replaced by a fluorescent dye, the signal suppression P and ThL (i.e., used for signal shaping and thresholding) can be achieved with a black hole quencher, and signal amplification and detection can be realized using another fluorescent probe [51, 52].

[51] Algar, W. R., Hildebrandt, N., Vogel, S. S. & Medintz, I. L. FRET as a biomolecular research tool — understanding its potential while avoiding pitfalls. *Nat. Methods* 16, 815–29 (2019).

[52] Fang, C., Huang, Y. & Zhao, Y. Review of FRET biosensing and its application in biomolecular detection. *Am. J. Transl. Res.* 15, 694–709 (2023).

III Response to Reviewer #3

I believe that the authors have adequately addressed my concerns from the original review.

Response:

We thank the reviewer for this positive feedback.

IV Response to Reviewer #4

I am impressed by the detailed and careful responses of the authors to the comments made by each of the reviewers including those made by me. The changes made by the authors provide clarity, additional information and are a genuine attempt to satisfy the reviewers and the original paper is much improved as a result.

Response:

We thank the reviewer for this positive feedback.